# Construction of a synthetic methodology-based library and its application in identifying a GIT/PIX protein–protein interaction inhibitor

Jing Gu[1,3], Rui-Kun Peng [1,3], Chun-Ling Guo[1,3], Meng Zhang[2], Jie Yang[1], Xiao Yan[1], Qian Zhou[1], Hongwei Li[1], Na Wang [1], Jinwei Zhu [2] ✉ & Qin Ouyang [1] ✉

In recent years, the flourishing of synthetic methodology studies has provided concise access to numerous molecules with new chemical space. These compounds form a large library with unique scaffolds, but their application in hit discovery is not systematically evaluated. In this work, we establish a synthetic methodology-based compound library (SMBL), integrated with compounds obtained from our synthetic researches, as well as their virtual derivatives in significantly larger scale. We screen the library and identify small-molecule inhibitors to interrupt the protein–protein interaction (PPI) of GIT1/β-Pix complex, an unrevealed target involved in gastric cancer metastasis. The inhibitor **14-5-18** with a spiro[bicyclo[2.2.1]heptane-2,3'-indolin]−2'-one scaffold, considerably retards gastric cancer metastasis in vitro and in vivo. Since the PPI targets are considered undruggable as they are hard to target, the successful application illustrates the structural specificity of SMBL, demonstrating its potential to be utilized as compound source for more challenging targets.

The primary task of medicinal chemistry is to invent therapeutic molecules for diseases. This invention is highly dependent on the coordinative development of two areas: discovery of new validated biological targets[1,2] and access to a large number of structurally diverse chemical compounds as screening source[3–5]. Therefore, nowadays compound libraries have been constructed on increasingly larger scales consisting of up to hundreds of millions of chemical entities[6] and have proved their potential in hit discovery for both conventional and novel biological targets.

Among these compound libraries, the natural product library is widely recognized as a large treasure base for drug development, which has consistently provided efficient pharmacotherapies for various diseases[7–9]. The high degree of molecular complexity and rigidity of natural products makes them more drug-like, especially for some special targets[10]. However, huge challenges still exist to developing natural products as new drug candidates: the limited source has hampered their large-scale production[11,12]. The molecular complexities and stereo-specificities have made the synthetic approach extremely laborious and costly[12–14].

Alternatively, organic synthetic methodology studies have afforded numerous compounds of diverse structures with high molecular complexities like natural products, many of which even share the same core scaffolds with some bioactive natural products[15,16]. Such natural product-like properties, plus their synthetic accessibility, make these

[1]Department of Medicinal Chemistry, College of Pharmacy, Third Military Medical University, 400038 Chongqing, China. [2]Bio-X Institutes, Key Laboratory for the Genetics of Developmental and Neuropsychiatric Disorders, Ministry of Education, Shanghai Jiao Tong University, Shanghai, China. [3]These authors contributed equally: Jing Gu, Rui-Kun Peng, Chun-Ling Guo. ✉e-mail: jinwei.zhu@sjtu.edu.cn; ouyangq@tmmu.edu.cn

compounds more suitable for hit discovery. Pitifully, only a very limited portion of them have been employed in further pharmacological research, although many preliminary evaluations have proved their bioactive potential[15,17–19]. Therefore, the construction of a synthetic methodology-based compound library would be beneficial for systematically studying their pharmaceutical potential[20,21]; and we assume that the special chemical space of the synthetic products would be particularly suitable for those challenging targets such as protein–protein interactions (PPIs)[22,23]. PPIs play critical roles in various cellular signal transductions, which makes them attractive drug targets in recent years[24–26]. However, due to their usually shallow and flexible binding interfaces, PPIs have been deemed challenging therapeutic targets, even undruggable[27,28].

In this work, we collect the products obtained from our series of synthetic methodology studies during the past few years and construct an in-house compound library (synthetic methodology-based library, the SMBL). Meanwhile, we encode the compounds and their possible derivatives to establish a large virtual library. From screening SMBL, we successfully identify two compounds capable of interrupting the interaction of GIT1/β-Pix complex, an undruggable PPI involved in gastric cancer metastasis. To our knowledge, no small-molecule inhibitor against this promising target has been reported, probably due to the limitation of conventional compound sources for screening PPI inhibitors. We further demonstrate that the newly identified compounds inhibit gastric cancer metastasis in vitro and in vivo.

## Results
### Construction of the synthetic methodology-based compound library
To construct the entity synthetic methodology-based library (SMBL-E), we collected the compounds synthesized using the published methodologies by our group since 2012 (Supplementary Fig. 1). The compounds were purified before collection, stored at −80 °C, also coded and numbered in a uniform manner. Within 10 years, more than 1600 compounds were collected. Scaffold types of the compounds were rich, including but not limited to indoles, quinolines, bridged, and spiro rings (Fig. 1a), which are frequently found in natural products but not easily found in commercial libraries. Notably, most of these compounds were constructed in an asymmetric manner, which endowed the library with stereochemical complexities.

On the other hand, large-scale virtual combinatorial compound libraries have clear advantages in drug discovery[3,4]. Therefore, based on the above entity library, a larger-scale virtual library (SMBL-V) was constructed. Not only the chemicals listed in the published papers but also more products designed using combinational chemistry and accessible via the reported methodologies were included in this virtual library. To construct such a virtual library, we first extracted the core scaffolds of the compounds; then analyzed the characteristics of these reported synthetic methods and identified the derivable site(s) of the compounds. Notably, to ensure all the designed compounds in this virtual library could be efficiently accessed, the groups for combination at each site were only limited to the scope proved by methodology studies. As shown in Fig. 1b, two compounds from one paper[29] were taken as examples to illustrate the process. For $R^1$, only aromatic groups were considered, while both heterocycles and alkyls were included for R according to the original research. For the virtual combination method, Legion module in Sybyl-X 2.0 was utilized. As a result, the virtual compound library (SMBL-V) containing over 14 million structures was constructed according to 144 papers during years 2008–2021 (Supplementary References). In theory, all the compounds in the library could be obtained quickly and conveniently through established methodologies. The construction of entity/virtual compound libraries has provided a valid basis with diverse structures for developing innovative hit compounds.

To prove the uniqueness of the new compound library (both SMBL-V and SMBL-E), we performed a similarity comparison[30,31] between our libraries and several commercially available libraries using a 2D fingerprint Tanimoto coefficient (Tc) calculation (Fig. 1c). Firstly, the similarity was compared versus the most popular commercial library *chembridge*. Both SMBL-V and SMBL-E showed small Tc values when compared to *chembridge* (Supplementary Table 1), suggesting low similarity ranges (Tc max approaching 1 stands for higher degrees of similarity of the two counterparts). On this basis, we chose two other commercial libraries—*targetmol* and *specs* to be compared with *chembridge*, which resulted in obviously higher Tc values. Furthermore, the comparison was also conducted between SMBL-V/E and *targetmol* or *specs*, which also resulted in low similarity (Fig. 1d). These comparisons suggested that the commercial libraries shared similar molecules or scaffolds, while our libraries were more different from them. These results also illustrated that most of the molecules in SMBL are unique in structure scaffolds and distinguishing from the molecules of the commercial libraries.

### 14-5-18 and 15-4-26 from SMBL as inhibitory small molecules for a functional PPI−GIT1/β-Pix
Next, we set to investigate whether SMBL has advantages for discovering hit compounds for those challenging targets. We focused on the GIT/PIX complex[32]. GIT (G protein-coupled receptor kinase-interacting) proteins, including GIT1 and GIT2, are GTPase-activating proteins (GAPs) for the ADP-ribosylation factor (Arf) family GTPases[33]. PIX (p21-activated kinase-interacting exchange factor) proteins, including α-Pix and β-Pix, are guanine nucleotide exchange factors (GEFs) for the Rho family GTPases (e.g., Rac1 and Cdc42)[34]. The GIT/PIX complex is recruited to the focal adhesion by adaptor protein paxillin and is involved in cell migration by activating downstream Rho GTPases[35,36]. Evidence has shown that GIT1/β-Pix function was highly correlated with cancer development and metastasis in a wide spectrum of human malignancies[32,37–39] including gastric cancer[40,41], but its exact functional roles were largely unknown.

We observed GIT1 expression was obviously elevated at the protein level in gastric cancer tissues (Supplementary Fig. 2a, b), as well as in several representative gastric cancer cell lines (Supplementary Fig. 2c). In MGC803 and MKN45, knockdown of *GIT1* significantly impeded the invasive behavior, while its overexpression apparently promoted the invasive ability (Supplementary Fig. 2d). Also, the amount of active Rac1 and Cdc42, the downstream factors of the GIT1/β-Pix complex, were significantly diminished when GIT1 was knocked down in both cell lines as expected (Supplementary Fig. 2e). According to published research, higher expression of GIT1 is strongly associated with a shorter survival time of patients with gastric cancer[41]. These results suggested that GIT1 acted as a vital regulating factor in gastric cancer metastasis with clinicopathological significance.

Next, we wanted to further confirm whether the GIT/PIX complex is involved in gastric cancer cell invasion. To address this question, we took advantage of the PIX-binding deficient mutation of GIT1. We previously showed that two conserved Leu residues (i.e., Leu271 and Leu279) were located at the interface of the GIT/PIX complex and critical for the complex assembly[42]. Consistent with our structural analysis, coimmunoprecipitation (co-IP) experiments showed that substitution of both Leu residues with Ala, GIT1$^{L271/279A}$ (hereafter referred to as GIT1$^{muts}$), largely abolished the GIT1/β-Pix interaction (Fig. 2a). In MKN45 cells, the impaired cell invasion ability caused by loss of GIT1 could be rescued by expression of wild type GIT1 but not the PIX-binding deficient GIT1$^{muts}$ (Fig. 2b). These results clearly demonstrated that the GIT1/β-Pix interaction is indispensable for gastric cell invasion, thus making the complex an attractive intervention target for gastric cancer metastasis. To this end, more effective pharmacotherapies targeting the GIT/PIX complex are urgently needed. However, based on the crystal structure of the GIT/PIX complex

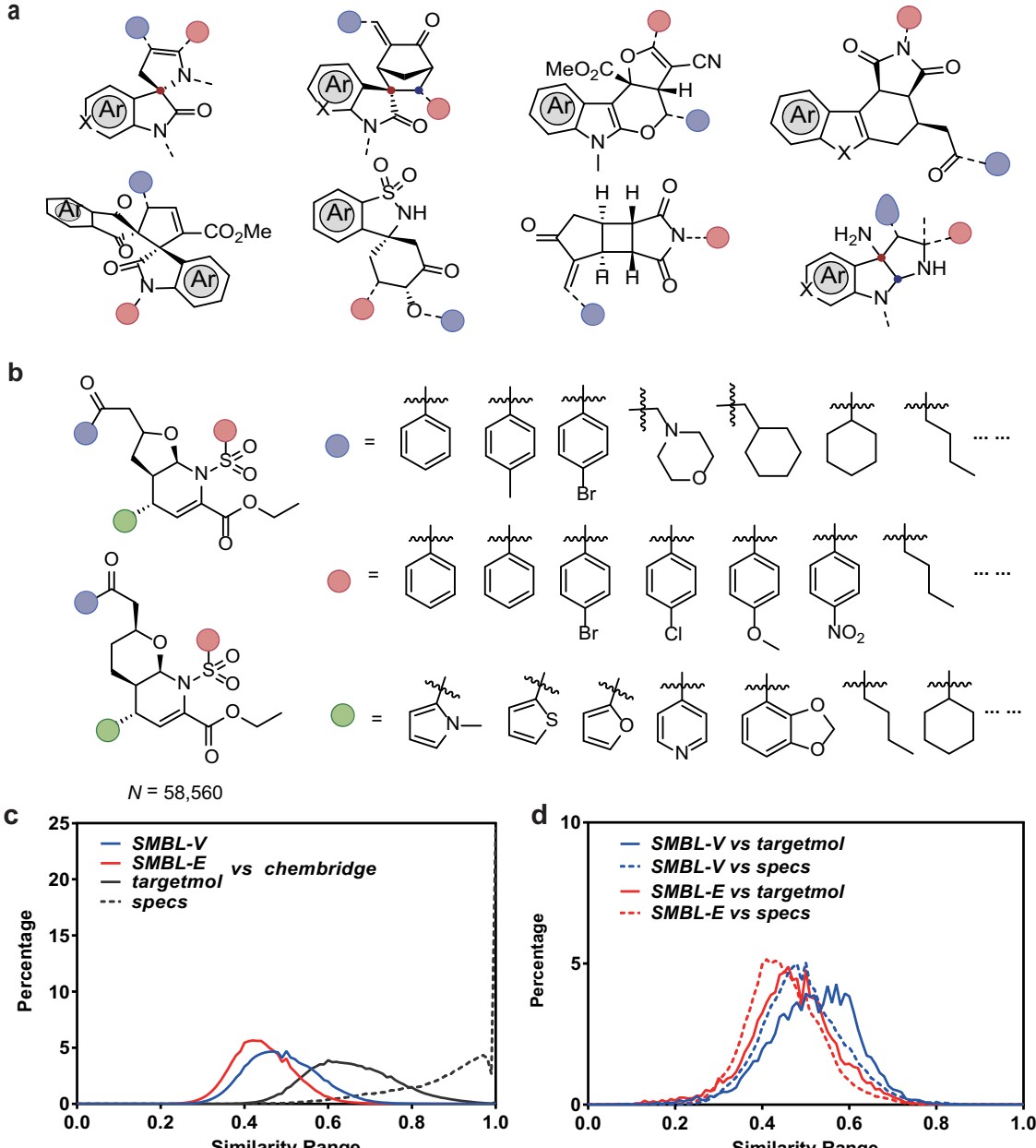

**Fig. 1 | Construction and analysis of SMBL. a** Representative scaffold of the entity library. The spheres show different substituents. **b** Derivation of the existing compounds to construct the virtual library. Two compounds from one paper (*Chin. J. Chem.* **2012**, *30*, 2669) were taken as examples to show the processes. The spheres show the derivable sites. **c**, **d** Similarity range between the SMBL and commercial libraries.

we solved recently[42], the interface of the complex was extremely shallow, which was hard to target by compounds in conventional libraries. Therefore, we intended to screen candidate inhibitory compounds capable of interfering with the GIT1/β-Pix interaction using our newly established SMBL.

The screening was performed at both virtual and entity levels. In the virtual screening, the docking model of the GIT1/β-Pix complex was constructed, and the binding pocket was generated based on the complex structure of GIT2/β-Pix that we solved recently (PDB ID: 6JMT)[42] (Fig. 2c). The virtual library (SMBL-V) was subjected to the Sybyl-X 2.0 program to perform high-throughput docking screening with GIT1 at the specific binding pockets. Consequently, the primary virtual screening resulted in 5659 compounds, cut-off at docking score > 5.0 (top 100 compounds in Source Data File). The Autodock and Sybyl-X 2.0 programs were used in combination for the second round of screening. Twenty top-ranked molecules were selected, and 9

compounds were synthesized (one structure for each scaffold) to perform further evaluation (N = 9) (see Source Data File for detailed structures and scores). For the entity library (SMBL-E) screening, a fluorescence polarization (FP)-binding assay[43,44] was established to evaluate the inhibitory abilities of the compounds on the GIT1/β-Pix interaction (Fig. 2d). Several representative compounds (N = 100) with different scaffolds were chosen for screening. Together with the 9 compounds selected from virtual screening, a total of 109 compounds were subjected to the FP binding assay to perform the final evaluation. Compound **15-4-26** from the virtual library and compound **14-5-18** from the entity library with the best binding affinities (*Ki* values) were selected as candidates (Fig. 2e and Supplementary Fig. 3).

Additionally, the commercial compound library *ChemDiv* was screened as a control. The 20 compounds with the best docking scores were selected, purchased, and further subjected to FP binding assay to test their inhibitory abilities. Unfortunately, this set of compounds

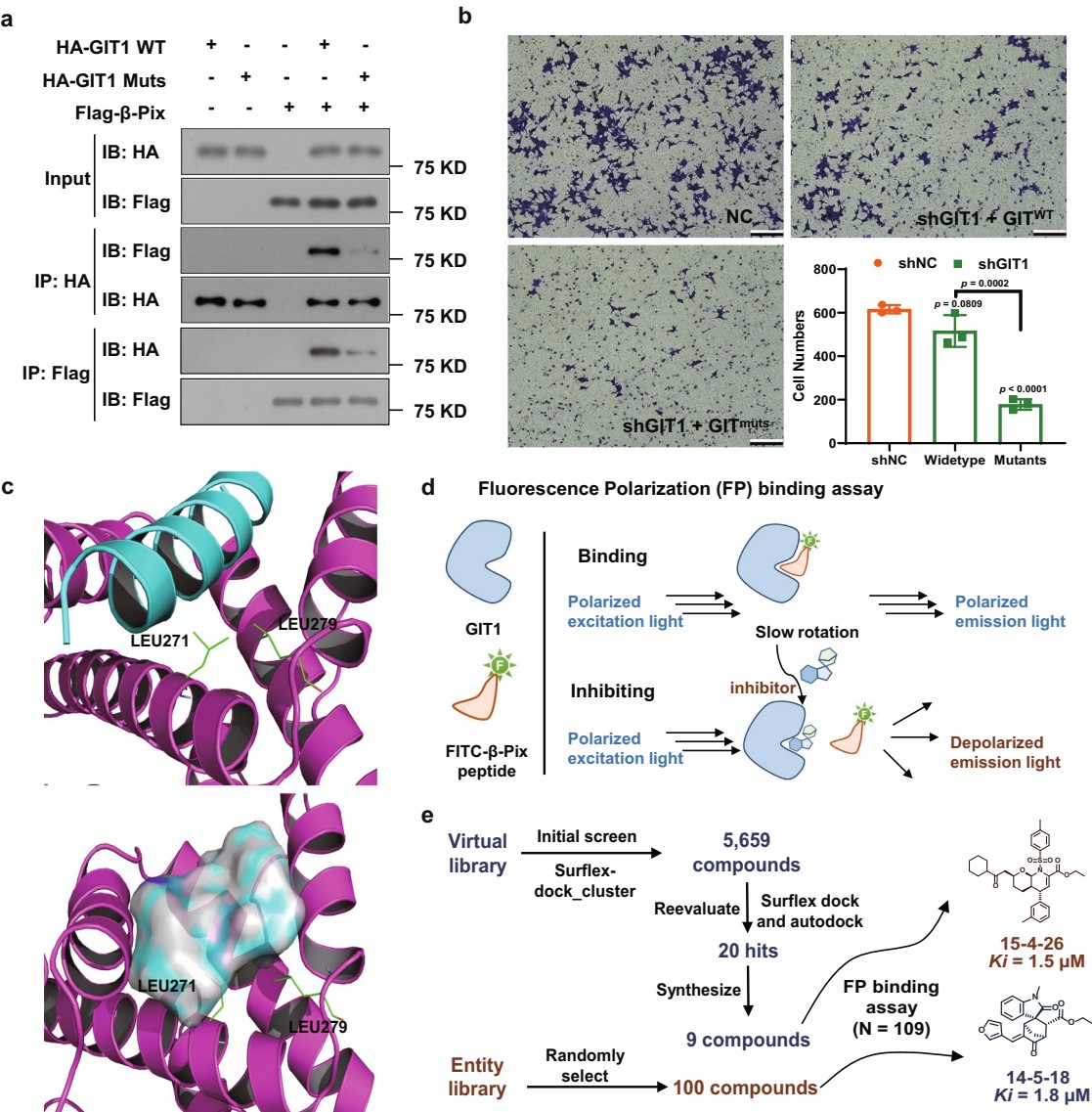

**Fig. 2 | Screening methods to identify 14-5-18 and 15-4-26 as inhibitory molecules for GIT1/β-Pix interaction. a** Co-immunoprecipitation (co-IP) experiment to detect the interaction between β-Pix and GIT1 in both the wild-type or mutant forms. HA-GIT1 (wide-type and mutant forms) and Flag-β-Pix were transfected into HEK-293T and the cell lysates were collected to perform the co-IP assay. The demonstrated blots were representatives from three independent experiments with similar results. **b** Transwell invasion assay to detect the invasion abilities of MKN45 cells with GIT1 wide-type or mutants (*n* = 3 biologically independent samples). Scale bar = 100 μm. **c** Cartoon model showing the interaction of GIT1 (magenta) and the key peptide β-Pix (cyan). Key residues Leu271 and Leu279 were

indicted in sticks (upper), and the pocket was generated for binding (down). **d** Schematic diagram of fluorescence polarization (FP) assay performed between GIT1 protein and FITC-β-Pix (5-FITC-Acp-ALEEDAQILKVIEAYCTSAKT). **e** Workflow of screening in both entity and virtual libraries to identify **14-5-18** and **15-4-26** as potential inhibitors. The *Ki* values were calculated according to IC$_{50}$ values of balance in the FP assay. Data are presented as mean values ± SD and error bars indicate SD. Statistical analysis: One-way ANOVA, Tukey's multiple comparisons tests, each group was compared with the other group. Source data are provided as a Source Data file.

provided no promising hits (Supplementary Table 2), further demonstrating the advantages of our SMBL during drug discovery for challenging biological targets.

## Compound 14-5-18 retarded gastric cancer metastasis by interrupting the GIT1/β-Pix interaction

We next investigated whether compounds **14-5-18** and **15-4-26** specifically perturbed the GIT1/β-Pix interaction in living cells. Firstly, we performed Cellular Thermal Shift Assay (CETSA) to preliminarily explore the target engagement of the compounds with GIT1 (Supplementary Fig. 4). CETSA relies on the fact that proteins are generally vulnerable to thermal conditions, while physical binding with other molecules could improve their stability[45]. In CETSA conditions, **14-5-18** could generally stabilize GIT1 in a dose-dependent manner ranging

from 10 to 100 μM under 65 °C, meanwhile, **15-4-26** was less efficient in stabilizing the protein when compound concentration increased, which may be due to its inferior solubility (Supplementary Fig. 4a). Therefore, further evaluations were only performed with **14-5-18**, and **15-4-26** was suspended for its poor potential for drug development. **14-5-18** exhibited an even better dose-dependent stabilizing effect on GIT1 when wide concentration ranges (0.5, 1.0, 5.0, 10, 25, 50, 100 μM) were set (Supplementary Fig. 4b). Similarly, under thermal conditions from 30 to 90 °C, **14-5-18** (50 μM) also considerably stabilized GIT1, especially at high temperatures (Supplementary Fig. 4c). To further confirm whether **14-5-18** directly bound to GIT1, we used biolayer interferometry (BLI) assay to measure their interaction. The BLI assay showed that **14-5-18** is directly bound to GIT1 with a $K_D$ value of 7.7 ± 0.1 μM (Fig. 3a and Supplementary Table 3).

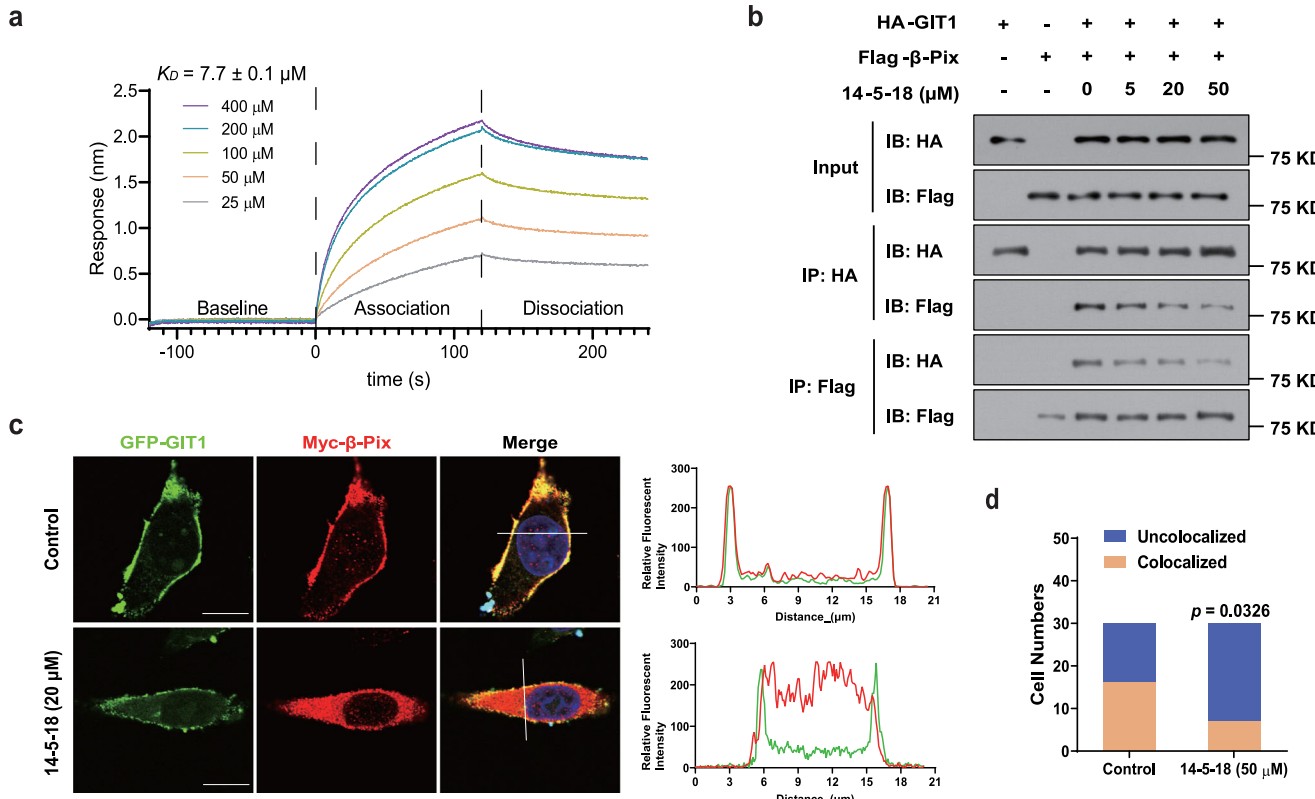

**Fig. 3 | Compound 14-5-18 interrupted the GIT1/β-Pix interaction. a** Kinetic measurement of the binding affinity of **14-5-18** to GIT1 by BLI assay. Concentrations ranging from 25 to 400 μM were shown with the real-time response for each step of the kinetic assay. **b** Co-IP assay to indicate the interaction of GIT1 and β-Pix and the inhibitory effects of **14-5-18**. HA-GIT1 and Flag-β-Pix were transfected into HEK-293T cells, and the cell lysates were collected to perform the co-IP assay at concentrations of 0, 5, 20, and 50 μM. The demonstrated blots were representatives from three independent experiments with similar results. **c** Fluorescence colocalization experiment to visualize the cell distribution of GIT1 and β-Pix with or without treatment of **14-5-18**. GFP-GIT1 (green) and Myc-β-Pix were expressed followed by treatment with a fluorescent secondary antibody (red) in the MGC803 cell line to construct the fluorescence system. The cells were observed under confocal microscopy after **14-5-18** was added at concentrations of 0 and 20 μM. Plot fluorescence intensities of the area marked by the white lines in the left panels are shown in the right graphs. Green and red curves represent the GIT1 and β-Pix, respectively. Scale bar = 10 μm. The demonstrated figures were representatives from three independent experiments with similar results. **d** The percentage of GIT1/β-Pix colocalized cells with or without treatment of **14-5-18**. The analysis was performed in $n = 30$ cells from three independent experiments. Statistical analysis: Two-sided Fisher's exact test (**d**). Source data are provided as a Source Data file.

We next studied whether **14-5-18** interfered with the GIT1/β-Pix interaction in living cells. Co-IP assay was performed in HEK293T cells expressing HA-tagged GIT1 and Flag-tagged β-Pix, where the interaction of GIT1 and β-Pix could be observed effectively. However, the addition of **14-5-18** gradually inhibited the interaction in a dose-dependent manner, ranging from 5 to 50 μM (Fig. 3b). Next, we wanted to investigate the effect of **14-5-18** on the cellular location of the GIT1/β-Pix complex. In cells overexpressing GFP-tagged GIT1 (green) and Myc-tagged β-Pix (red), we observed obvious co-localization of GIT1 and β-Pix at the cell periphery by confocal microscopy (Fig. 3c). Consistent with the binding assays, the co-localization of two proteins was largely suppressed when **14-5-18** was added to the cells (Fig. 3c, d). Taken together, these results demonstrated that **14-5-18** could bind to GIT1 and effectively inhibit the interaction between GIT1 and β-Pix in living cells.

Since the candidate compounds could interfere with the GIT1/β-Pix interaction, we next tested whether they could affect gastric cancer cell invasion. In the in vitro invasion experiment using a transwell assay, **14-5-18** exhibited significant inhibitory effects on the invasiveness of both MKN45 and MGC803 cells at 10 μM, with a better inhibition effect at the concentration of 50 μM (Fig. 4a and Supplementary Fig. 5a). This dose-dependent inhibitory effect proved that the compound not only disrupted GIT1/β-Pix interaction at the molecular level but also impeded its function in the cellular level. Notably, the compounds showed a very slight influence on cell

viability even at high doses, detected by CCK8 assay and CellTiterGlo assay (Supplementary Fig. 5b–e), partially suggesting the target specificity of these compounds. As described earlier, disruption of the GIT/PIX interaction could impact the activation of the downstream Rho GTPase Rac1 and Cdc42. Therefore, we reasoned that **14-5-18** treatment would lead to a reduction of the level of active Rac1 and Cdc42 in gastric cancer cells. As expected, GTP-Rac1 and GTP-Cdc42 were significantly suppressed by **14-5-18** in a dose-dependent manner (ranging from 5 to 50 μM), while total Rac1 and Cdc42 expression were unaffected (Fig. 4b).

Furthermore, we detected whether the compounds exhibited inhibitory effects on cancer metastasis in vivo. Luciferase-expressing MGC803 cells were constructed and injected intravenously via the tail vein of nude mice to establish a metastasis model[46]. After luminescence observed in the lungs, **14-5-18** was administered intragastrically at doses of 0, 10, and 30 mg/kg (Supplementary Fig. 6a). The luminescence intensity in the lungs was significantly reduced in a dose-dependent manner when treated with **14-5-18** (Fig. 4c). Consistently, H & E staining of the lung tissues also showed that the metastatic foci of the treated groups were obviously smaller in size and number (Supplementary Fig. 6c). The safety status of **14-5-18** was also reliable, as the body weight curve demonstrated that drug administration brought no extra burden to mice other than the tumor itself (Supplementary Fig. 6b).

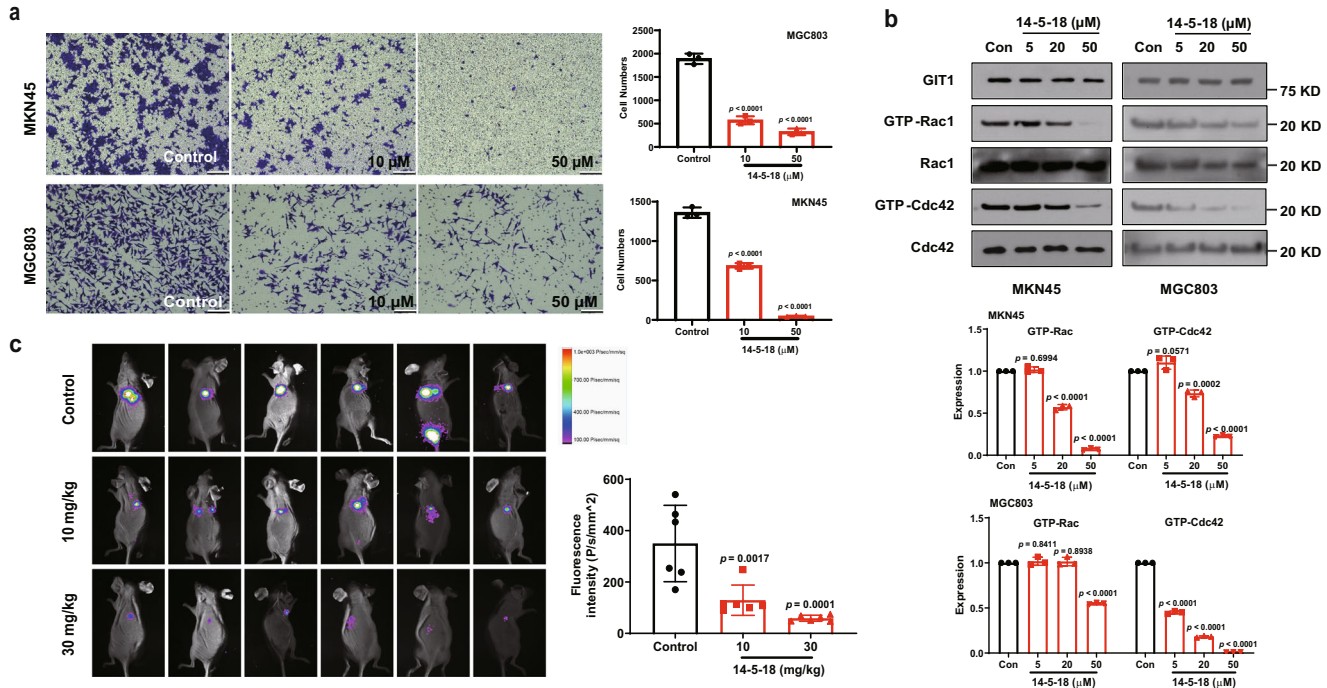

**Fig. 4 | Compound 14-5-18 inhibited invasion of gastric cancer cells in vitro and in vivo. a** Transwell assays were employed to detect the anti-invasion abilities of **14-5-18** (10 and 50 μM) in the MGC803 (left) and MKN45 (right) cell lines for 24 h. The right columns show quantification, $n = 3$ biologically independent samples. Scale bar = 100 μm. **b** Expression levels of total Rac1/Cdc42 and the activated forms GTP-Rac1/Cdc42 were detected after **14-5-18** was added to MKN45 and MGC803 at concentrations of 5, 20 and 50 μM for 24 h, $n = 3$ biologically independent samples. **c** In vivo imaging of nude mice after intravenous injection of luciferase-expressing MGC803 cells. Compound **14-5-18** were administrated at the indicated dosages by gavage once a day for 18 days ($n = 6$ biologically independent animals), and solvent DMSO (2%) + Tween 20 (5%) in water was used as control. Otherwise noted, DMSO was used as solvent control. Data are presented as mean values ± SD, and error bars indicate SD. Statistical analysis: One-way ANOVA, Dunnett's multiple-comparisons test (**a–c**, each group was compared with the control group). Source data are provided as a Source Data file.

## Molecular basis of inhibitory effect of 14-5-18 on the GIT1/β-Pix interaction

We next tried to look into the underlying mechanism of the inhibitory effect of **14-5-18** on the GIT1/β-Pix interaction. The molecular docking of GIT1 and **14-5-18** was first performed to analyze the binding interactions. To generate a more rational binding interface, the energies of the top three models with the highest docking scores were calculated by molecular dynamics simulations and MM/GBSA approach (Supplementary Fig. 7 and Supplementary Table 4). The one with the lowest binding energy (−32.0 ± 3.8) kcal/mol was selected (shown in Fig. 5a). According to the per-residue energy decomposition results, we found that several residues in the GIT1 binding pocket generated an interacting force with **14-5-18**, including Leu271 and Leu279 (with −0.24/−2.0 kcal/mol energy decomposition) (Fig. 5a), suggesting that **14-5-18** may compete with β-Pix when bound to these sites. Moreover, we tried to determine whether the mutants of GIT1 (GIT1[Mut1]: GIT1[L271A]; GIT1[Mut2]: GIT1[L279A]) influenced the drug-induced inhibition of gastric cancer cell invasion and Rac1 activation. We reasoned that if **14-5-18** targeted GIT1/β-Pix interaction at the site including Leu271 and Leu279 of GIT1, **14-5-18** would show no notable effect on cell invasion in the cells expressing the respective GIT1 mutants. In the knock-down cell line MKN45-sh*GIT1*, expression of GIT1[WT] successfully rescued the cell invasion ability (Fig. 5b and Supplementary Fig. 8a). However, the rescue effect was reversed in the presence of **14-5-18**. In sharp contrast, MKN45-sh*GIT1* cells expressing GIT1[Mut1] or GIT1[Mut2] remained insensitive to **14-5-18** treatment (Fig. 5b). In other words, **14-5-18** could not inhibit the cell invasion when Leu271 or Leu279 of GIT1 was mutated. In accordance with this result, in the MKN45-sh*GIT1* cell line, Rac1 was activated when GIT1[WT] was expressed and such activation was later inhibited in the presence of **14-5-18**, whereas the activity of Rac1 did not change when expressed with GIT1[Mut1] or GIT1[Mut2], with or without

**14-5-18** (Fig. 5c and Supplementary Fig. 8b). Co-IP experiments was also performed with mutant GIT1. As expected, **14-5-18** lost its inhibitory ability when Leu271 and Leu279 were mutated, as determined by immunoprecipitation of GIT1 to β-Pix (Supplementary Fig. 8c). Together, these results suggested that **14-5-18** targeted GIT1/β-Pix interaction at the sites including residues Leu271 and Leu279 of GIT1.

## Discussion

Chemical libraries are important for drug discovery. The skeleton diversity, synthetic accessibility, and library scale are critical to the speed and quality in the development of lead compounds for small-molecule drugs[47,48]. Compound libraries could be mainly classified into four types: (1) million-level entity libraries, which are large, accessible but costly, and usually have low structural complexity; (2) DNA-coding libraries, whose diversities are limited due to the limitation of reaction types; (3) virtual combinatorial libraries, which are randomly generated in silico thus usually lack synthetic accessibility; (4) natural product libraries with diverse and drug-like chemical space but limited sources and complex synthesis. Therefore, to develop new compound libraries with high degrees of structural diversity and synthetic feasibility is of great significance.

The SMBL established in this manuscript meets the requirements of synthetic feasibility and structural diversity. As far as we know, an entity/virtual library based on synthetic methodologies similar to SMBL has not been reported yet. The library contains two components: one is the entity library (SMBL-E), consisting of products collected from the published articles (about 1600 compounds); the other one is the virtual library (SMBL-V), including over 14 million compounds derived from the entities. Although our virtual library is not as large as some combinational libraries[3,4,6], all the compounds were meticulously designed according to the reported substrate scope of

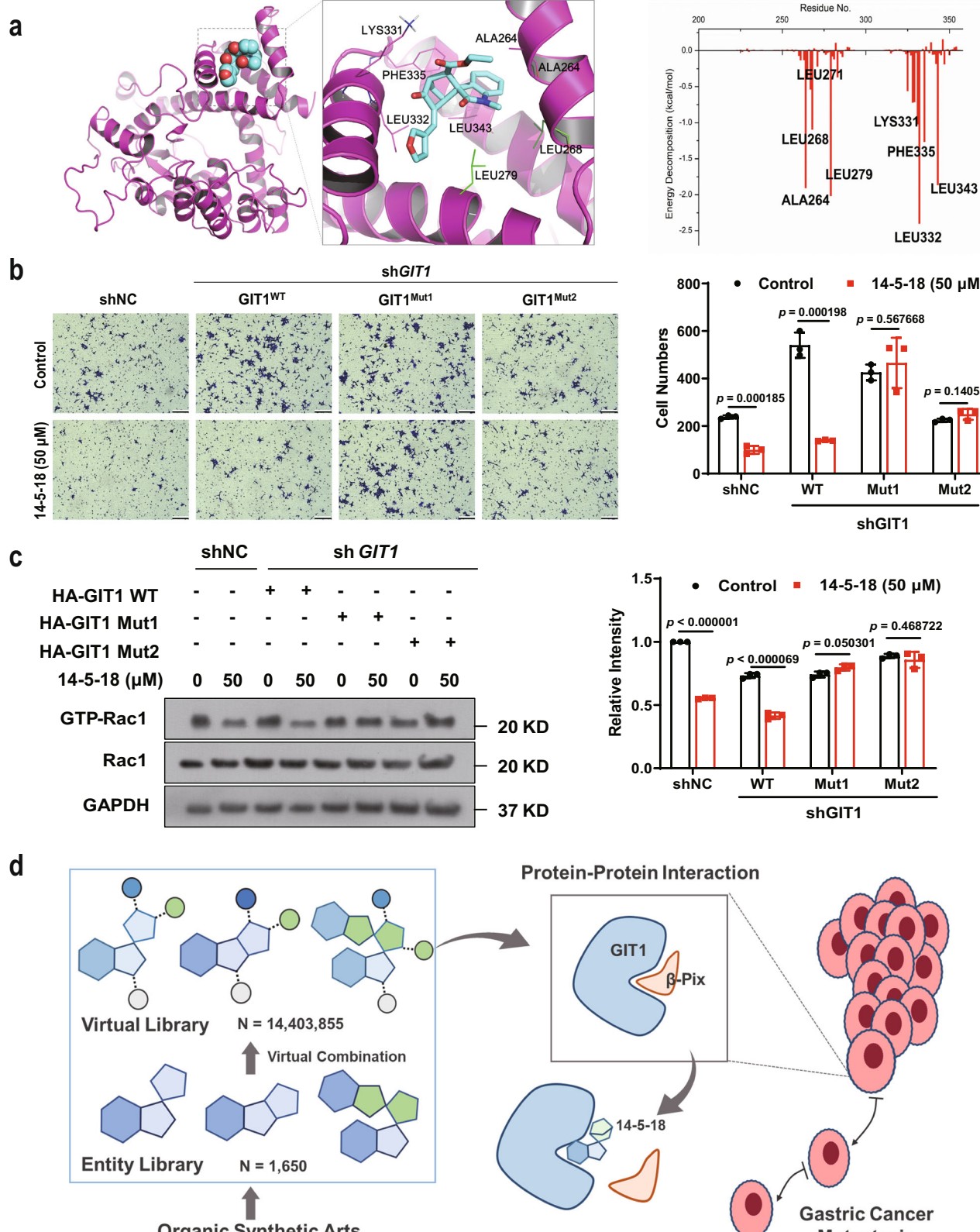

**Fig. 5 | Compound 14-5-18 targets the GIT1/β-Pix interaction at GIT1 residues Leu271 and Leu279. a** Cartoon model showing the binding pose of GIT1 (magenta) and **14-5-18** (cyan). Magnification of the binding pocket (middle) indicates the key residues (in lines) adjacent to **14-5-18** (in sticks). The graph (right) shows the energy decomposition of each adjacent residue. **b** Transwell invasion assay to detect the invasion abilities of MKN45 cells with wide-type or mutant GIT1, with or without

**14-5-18** (50 μM) for 24 h ($n = 3$ biologically independent samples). Scale bar = 100 μm. **c** The cellular activity of Rac1 in MKN45 cells with wide-type or mutant GIT1 with or without treatment of **14-5-18** (50 μM) for 24 h ($n = 3$ biological independent samples). **d** The graphic abstract. Otherwise noted, DMSO was used as solvent control. Data are presented as mean values ± SD, and error bars indicate SD. Statistical analysis: unpaired two-tailed $t$-tests. Source data are provided as a Source Data file.

the methodology studies. Theoretically, every in-library compound could be obtained via concise synthesis.

Compared with the conventional commercial libraries, in SMBL we could find a higher percentage of unique and complex structure types, including various polycyclic systems, heterocycle systems, and bridged, fused, or spiro rings. The comparison between SMBL and commercial libraries including *chembridge*, *specs,* and *targetmol* showed low similarity, suggesting that our library is unique in structure types. The uniqueness would offer new chemical space for hit discovery. Moreover, most of the compounds in SMBL are synthesized in asymmetric manners, containing one or several stereogenic centers. Therefore, the other major superiority of this library is the complexity of the compound structures. In this manuscript, the compound source was limited to published research of our group, the number, and diversity of the library could be easily expanded when more organic synthetic methodologies were involved.

To prove whether our library has some superiorities over conventional libraries, we screened for hit compounds targeting some challenging undruggable targets such as PPIs[22,23,27,28]. PPIs are known as 'undruggable' because of their flat and flexible interfaces. But PPIs mediate a huge array of cellular functions, which made their modulators in urgent need. Most existing PPI inhibitors are peptides, and small molecules targeting PPIs are quite rare[49], partially due to the limitation of conventional libraries. The successful identification of the **14-5-18** targeting GIT1/β-Pix interaction—a functional PPI, could partially prove that our library has certain advantages in lead discovery for some challenging biological targets.

We focused on GIT1/β-Pix interaction for its potential regulatory function in gastric cancer metastasis. Gastric cancer is one of the most common malignancies and has become the third leading cause of cancer deaths in China[50]. Due to the high invasiveness of gastric cancer, remote metastasis is the main cause of patients' death. In this manuscript, our study suggested that GIT1 expression is upregulated and its interaction with β-Pix promotes gastric cancer metastasis. Thus, inhibiting GIT1/β-Pix interaction would possess the potential for gastric cancer therapy. Previous research has pointed out that an endogenous protein, Naa10p, could act as an inhibitor of GIT1/β-Pix binding and suppress cell invasion in lung cancer[38,51]. Although exerting high activity, proteins such as Naa10p are somewhat unbefitting for therapeutic use due to their intrinsic properties, such as low stability and immunogenicity[52]. To the best of our knowledge, no inhibitor other than Naa10p has been reported previously. We attempted to find small-molecule inhibitor(s) for this challenging 'undruggable' target—GIT1/β-Pix. The commercially available libraries were first screened, but no hits were identified. Fortunately, our self-constructed SMBL provided promising molecules, both **14-5-18** (from the entity library) and **15-4-26** (from the virtual library) are capable of binding GIT1 to interfere with the interaction. We also identified **14-5-18** as a promising lead to inhibit metastasis in vitro and in vivo.

The molecule **14-5-18** was from our previous paper[53], with a complex scaffold—spiro[bicyclo[2.2.1]heptane-2,3′-indolin]-2′-one, which includes continuous chiral centers and spiro-bridged ring systems. We searched the scaffold of **14-5-18** in *ChemBL* (https://www.ebi.ac.uk/chembl/), a manually curated online database of bioactive molecules including 2,331,700 compounds with distinct scaffolds, and found no similar structure even in such a multifarious database. These analyses highlight the uniqueness of the scaffold.

As a small-molecule compound, **14-5-18** is more suitable for drug development, considering its good stability and low production cost compared with known ligands such as proteins and peptides[54]. Moreover, **14-5-18** was orally administrated to model animals and exerted adequate therapeutic efficacy, suggesting that the compound has fair bioavailability. Notably, while inhibiting cell invasion, the hit compounds showed a very slight influence on cell survival, suggesting that they might have reliable drug safety as therapeutics. This property is

rarely found among small-molecule anticancer agents, partially owing to the good target specificity of the molecules. Therefore, we assume that these compounds, or other strategies targeting GIT1/β-Pix, have the potential to prevent post-surgery metastasis, which is a frequent occurrence in cancer patients that have no accepted interventive therapeutics.

In addition to cancer cell migration and invasion, the GIT/PIX complex also plays key roles in the nervous system and immune system, mediating vital physiological functions[32]. Dysregulation of GIT/PIX function could cause severe neuropsychopathic diseases, such as Huntington's disease[55] and attention deficit hyperactivity disorder (ADHD)[56], immunological diseases such as Crohn's disease[57], and susceptibility to tissue injury[58]. Most mechanisms of the abovementioned physiological or pathological processes are still unclear. Therefore, **14-5-18** would also act as a manipulating tool compound to probe the cellular processes involving the GIT/PIX complex.

Admittedly, the research toward **14-5-18** was far from sufficient, and major problems remain to be settled in further study. One is that more in vivo experiments should be carried out to clarify its ADME properties for optimal administration, as well as to prove that the therapeutic effects are mediated by targeting GIT1/β-Pix interaction. The other aspect is that structural modification is needed to obtain derivatives with better activities, since the effective concentration of **14-5-18** is still high at present.

In conclusion, utilizing our recently established synthetic methodology-based natural product-like library, we successfully identified two promising inhibitors, targeting the GIT1/β-Pix interaction. Since PPIs are challenging targets for conventional compound libraries, we assume that the successful identification of these two hits demonstrated the potential of this library as a compound source for PPI inhibitor screening (Fig. 5d). The identification of hit compounds against such challenging and novel targets is of significant and growing interest to the broader drug discovery community. We hope this work would encourage more chemists in the organic synthetic methodology field to collect their products and use them for hit discovery, promoting the connection between the well-developed organic synthetic arts and drug development[59,60].

## Methods
All research complies with relevant ethical regulations. The use of the tissue in immunohistochemical (IHC) staining for tissue microarray analysis has been approved by the institutional Ethics Committees of Shanghai Outdo Biotech (China). The animal experiments were approved by the Laboratory Animal Welfare and Ethics Committee of Third Military Medical University under approved protocol AMUWEC20200959.

### Compound sources
Commercial compounds were purchased from *ChemDiv*. Compounds from the entity library were obtained from stockage or synthesized following the articles if a larger scale was needed (Supplementary Methods). All compounds were confirmed by $^1$H-NMR and consistent with the original papers. New compounds from the virtual library were synthesized according to the reported methodologies and identified by $^1$H-NMR, $^{13}$C-NMR, and HRMS. All final compounds were purified to >95% purity before use, determined by HPLC analysis.

### Fluorescence polarization (FP) assay
*GIT1 protein purification*: Purification of GIT1 1−370 was performed following a published protocol[42]. Briefly, the coding sequence of mouse GIT1 aa. 1−370 (GenBank: NM_001004144.1) was cloned into a pET32M3C (with an N-terminal Trx-His$_6$ tag) vector. The recombinant protein was expressed in *Escherichia coli* BL21 (DE3) cells at 16 °C for 18 h induced by the isopropyl-β-D-thiogalactoside (IPTG) at a final concentration of 0.2 mM. The N-terminal Trx-His$_6$-tagged GIT1 1−370

was purified by Ni$^{2+}$-NTA agarose affinity chromatography, followed by a Superdex-200 26/60 size-exclusion chromatography (GE Healthcare, Cytiva). To remove the Trx-His$_6$ tag, GIT1 protein was cleaved by HRV-3C protease at 4 °C overnight and then purified by another step of SEC purification. The β-Pix-positive peptide (sequence: ALEEDAQILKVIEAYCTSAKT) and fluorescent peptide (sequence: FITC-(Acp)-ALEEDAQILKVIEAYCTSAKT) were synthesized by China peptide (Shanghai, China). FP assay was used to monitor interactions between GIT1 and the compounds. For each assay, fresh protein stocks of GIT1 were thawed. Assay buffer contained 100 mM NaCl, 50 mM Tris pH 7.5. All the experiments were prepared in duplicates. The fluorescence intensities, parallel and perpendicular to the plane of excitation, were determined in Corning black 96-well NBS assay plates at room temperature. Fluorescence polarization was determined using a Multi-Mode Detection Platform (SpectraMax Paradigm, Molecular Devices, USA) with 485 nm excitation and 535 nm emission filters, the values were expressed in millipolarization units (mP). The binding affinity of the fluorescent peptide toward GIT1 was first determined via titration. For this purpose, 100 nM fluorescent peptide was contacted with serial dilutions of GIT1 (concentration ranging from 0.01 to 10 μM) in a final volume of 80 μL. $K_D$ value was stimulated from FP values and protein concentrations using GraphPad Prism 8.0 software. Competition binding assay was performed using 100 nM fluorescent peptide and optimal protein concentration (2.0 μM) for the measurement was calculated based on determined $K_D$. Tested compounds were prepared as serial dilutions (0.1–500 μM) in DMSO. β-Pix positive peptide (25 μM) was used as a positive control. IC$_{50}$ values of the compounds were stimulated according to the FP values and compound concentrations, using GraphPad Prism 8.0 software. $Ki$ values were calculated using the method reported by Wang et al.[61].

## Cell culture

Cell lines MKN45, MGC803, HEK-293T, BGC823, SGC7901, and GES-01 were constructed and provided by the Department of Gastroenterology, Xinqiao Hospital, Third Military Medical University (Chongqing, China), and MKN45, MGC803, HEK-293T, BGC823 were cultured in DMEM (Thermo, USA) medium containing 10% fetal bovine serum (Lonsera, URY), 10,000 U/mL penicillin, 10,000 μg/mL streptomycin (Thermo, USA). SGC7901, GES-01 were cultured in RPMI-1640 (Thermo, USA) medium containing 10% fetal bovine serum (Lonsera, URY), 100 U/mL penicillin, and 100 μg/mL streptomycin (Thermo, USA). Cells were incubated at 37 °C under a 5% CO$_2$ atmosphere and passaged when the cell confluence reached 80%.

## Immunohistochemical (IHC) staining for tissue microarray analysis

The tissue microarray (chip No. HStmA180Su11), which contained samples from 89 cases of human gastric carcinoma, was obtained from Shanghai Outdo Biotech (China). Informed consent was obtained from all subjects. GIT1 staining was scored using a positivity rate, where a score of >90% was classified as a high expression.

## Anti-proliferative assay

The anti-proliferative activity of **14-5-18** and **15-4-26** against MKN45 and MGC803 cell lines were evaluated using cell counting kit-8 assay and CellTiterGlo (CTG) assay. For the CCK-8, cells were plated on 96-well plates at a density of 5000 cells/well and cultured overnight. After the cells have reattached the plate, administrate the compounds at concentrations of 1.25, 2.5, 5.0, 10, 20, 40, 80, 100 μM and incubate for 24 h. Then 10% CCK-8 reagent (Beyotime, China) was added and incubated for 2 h. The absorbance at 450 nm was then read on a Multi-Mode Detection Platform (SpectraMax Paradigm, Molecular Devices, USA) to calculate the cell viability. CTG which detects the ATP level was also performed. Cells were plated on 96-well opaque-walled plates at a density of 5000 cells/well and cultured overnight. After the cells have

reattached the plate, administrate the compounds at concentrations of 1.25, 2.5, 5.0, 10, 20, 40, 80, and 100 μM and incubate for 24 h. Then 50% CTG reagent (Promega, USA) was added, vibrated for 2 min, and incubated for 10 min. The luminescence intensity was then read on a Multi-Mode Detection Platform (SpectraMax Paradigm, Molecular Devices, USA) to calculate the ATP level.

## Western blotting assay

Cells were collected and lysed using lysis buffer (Beyotime, China) with the addition of proteinase inhibitor PMSF and a cocktail phosphatase inhibitor (Beyotime, China). Centrifuge under 16,200×$g$ at 4 °C and remove the precipitated pellets, total proteins in the supernatants were quantified using the BCA kit (Beyotime, China). Loading buffer 5× was added and the protein was denatured at 95 °C for 10 min. Prepared samples containing 30 μg total protein were then subjected to 10% SDS–PAGE (Bio-Rad, USA), and transferred to a PVDF membrane (Bio-Rad, USA). The membranes were incubated with indicated primary antibodies including GIT1 (Cell Signaling Technology, CST#2919, 1:1000 dilution), HA-tag (CST#3724, 1:2000 dilution), Flag-tag (CST#14793, 1:1000 dilution), GAPDH (CST#2118, 1:1000 dilution), Rac1 (CST#2465, 1:1000 dilution), Cdc42 (CST#2462, 1:1000 dilution) overnight at 4 °C, followed by HRP-labeled secondary antibody (Bio-Rad #1706515/1706516, 1:3000 dilution) for 2 h at room temperature. Finally, the protein bands were visualized using a chemiluminescence kit (Bio-Rad, USA). Quantification of the bands was figured using ImageJ software.

## Transwell assay

Transwell assays were performed using an 8-μm-pore-size bottom filter chamber (Corning, USA) coated with Matrigel (300 μL/Ml, Corning, USA). Cells (5 × 10$^4$) were starved in serum-free DMEM medium overnight and seeded in 200 μL of serum-free DMEM medium onto the upper chamber. The lower compartment was filled with DMEM medium supplemented with 10% serum. After incubating at 37 °C for 24 h, the cells that had migrated to the lower surface of the filter were fixed and stained with 0.1% crystal violet. Cells remaining on the upper surface of the filter were wiped off with a cotton swab, and then the stained cells were counted under ×10 microscopic fields.

## Transfection

Plasmid DNA was transfected at a 1:3 ratio using PEI buffer into overnight exponentially grown cells in 10 cm cell culture dishes. Transfected cells were cultured in the medium and incubated for 24-48 h at 37 °C with 5% CO$_2$.

## GIT1 and β-Pix expression

To generate the GIT1 and β-Pix expression vector, a full-length DNA sequence of human GIT1 and β-Pix complementary DNA (cDNA) was cloned into the expression vector pcDNA3.1-HA-C (Youbio Inc., #V2187) and pcDNA3.1-3xFlag-C (Youbio Inc., # VT9221), respectively, which were fused to a C-terminal HA or 3xFlag tag. The GFP-GIT1 and Myc-β-Pix expressing plasmids as well as mutants of GIT1 were generated following procedures established before[42]. For GIT1$^{Mut1}$, the sequence LSNRLFEEL was mutated to ASNRLFEEL, for GIT1$^{Mut2}$, LSNRLFEEL was mutated to LSNRLFEEA, while for GIT1$^{Muts}$, LSNRLFEEL was mutated to ASNRLFEEA. All constructs were checked by DNA sequencing, the primers are listed in a Source Data File. The vectors were transfected into HEK-293T or gastric cancer cells according to the abovementioned transfection methods, and expression levels were confirmed by Western blotting.

## GIT1 knockdown

The lentiviral short hairpin (sh)RNA targeting GIT1 was designed and constructed by Genechem Co. Shanghai (#90249-1 for sh*GIT1*-a, and #90250-1 for sh*GIT1*-b). For lentivirus infection, MKN45 and MGC803

cells were divided into a negative control group (shNC), and a test group (sh*GIT1*-a, sh*GIT1*-b). Cells were seeded into a six-well plate and transduced with lentiviral particles at a multiplicity of infection of 10. The lentiviral vector contained a puromycin-resistant fragment, thus puromycin was used to screen successfully transfected cells. 24 h post-infection, cells were harvested for Western blotting to detect GIT1 knockdown.

## Cellular thermal shift assay (CETSA)

CETSA was performed following published protocols[62,63]. For the temperature-dependent CETSA, 50 μL of lysates (3 mg/mL) from MGC803 cells were incubated with 50 μM of compounds at temperature points of 30, 40, 50, 60, 70, 80, 90 °C for 5 min. The samples were centrifuged at 20,000×$g$ for 10 min at 4 °C to separate the supernatant and pallet. 12 μL of the supernatant was mixed with 3 μL of 5× loading buffer and then separated on a 10% SDS–PAGE for immunoblotting analysis of GIT1. For the dose-dependent thermal shift assay, 50 μL of lysates (3 mg/mL) were incubated with compounds at concentration points of 0, 10, 20, 30, 40, 50, 60, 70, 80, 90, 100 μM or 1.0, 5.0, 10, 25, 50, 100 μM under 65 °C for 5 min. The samples were then subjected to immunoblotting analysis of GIT1 as described above.

## Biolayer interferometry (BLI) assay

The direct interaction between **14-5-18** and GIT1 was detected using ForteBio Octet RED (Sartorius, Germany). GIT1 (1 mg/mL) was biotin-labeled using an EZ-Link™ NHS-PEG4-Biotin kit (#A39259, Thermo-Fisher, USA) following the manufacturer's protocols. 0.02% PBST was used as the assay buffer. GIT1 concentration was adjusted to 100 μg/mL, and the compound was diluted to 400, 200, 100, 50.0, and 25.0 μM, respectively, a solvent control (0 μM) was also prepared. Super Streptavidin (SSA) biosensors (Sartorius, Germany) were pre-hydrated for 10 min before use. Then the procedures were performed on the Octet BLI Discovery platform to detect the binding, which consisted of a baseline step (60 s), loading (300 s), the second baseline step (120 s), association (120 s), and dissociation (120 s). The response was recorded over time. Reference sensors without GIT1 loading served as background controls for each concentration. Curves were fit to a 1:1 interaction model, and kinetic analysis was performed using Octet Analysis Studio software (Sartorius, Germany) to calculate the $K_D$ value.

## Co-immunoprecipitation (Co-IP)

About 1–2 × 10^6 sorted HEK-293T cells stably expressing HA-GIT1, Flag-β-Pix, HA-GIT1^Mut1, or HA-GIT1^Mut2 were grown individually in 10 cm cell culture dishes in Medium HEK293T at 37 °C with 5% CO$_2$. The culture medium was discarded, and the cells were washed with ice-cold PBS. After discarding the PBS, cells were lysed in 400 μL ice-cold lysis buffer (for co-IP) with the addition of proteinase inhibitor (Beyotime, China) for 30 min at 4 °C with gentle shaking. Centrifuge under 16,200×$g$ for 15 min and remove the precipitated pellets, total proteins in the supernatants were quantified using the BCA kit (Beyotime, China). Subsequently, 10 μL of monoclonal anti-HA-agarose antibody (Sigma, USA) or monoclonal anti-Flag-agarose antibody was added to 200 μg pre-cleared lysate and incubated overnight at 4 °C with rotation. Centrifuge under 16,200×$g$ for 5 min and remove the supernatant, then wash the beads with NETN buffer, bound proteins were released by incubation in 30 μL 4× SDS–PAGE reducing protein loading buffer (Bio-Rad, USA) at 95 °C for 10 min and centrifuged for 1 min, 16,200×$g$. The supernatant was collected and either used directly for SDS gel electrophoresis or stored at −80 °C, and then separated on 8 % SDS–PAGE for Western blotting analysis.

## Immunofluorescence microscopy

MGC803 cells were cultured on slides in 24-well plates, and plasmids expressing GFP-GIT1 and Myc-β-Pix were transfected for 24 h. Then cells were incubated with 20 μM **14-5-18** for another 24 h. Next, the cells were fixed on the slide for 30 min with 4% paraformaldehyde, permeabilized with 0.1% TritonX-100 for 5 min, then blocked with goat serum for 40 min. The cells were then washed three times in PBS and stained with anti-Myc primary antibody (CST#2278) overnight at 4 °C, followed by Cy3-fluorescent secondary antibody (Beyotime#A0516, 1:800 dilution) for 2 h at room temperature. Fluorescent images were obtained using a laser scanning confocal microscope (OLYMPUS FLUOVIEW 1000), handled using Zen software and the plot fluorescence intensity profiles were obtained using ImageJ.

## Active Rac1/Cdc42 detection

Active Rac1/Cdc42 Detection assays were performed following the protocols of Active Rho/Cdc42 Detection Kit (Cell Signaling Technology, USA). Cells were seeded into a six-well plate and incubated overnight before transfection or treatment with compounds. The culture medium was discarded, and the cells were washed with ice-cold PBS. Add 0.5 mL ice-cold 1x Lysis/Binding/Wash buffer plus 1 mM PMSF to each plate (10 cm in diameter). Cells were lysed for 5 min at 4 °C with gentle shaking. Centrifuge under 16,200×$g$ for 15 min and remove the precipitated pellets, total proteins in the supernatants were quantified using the BCA kit (Beyotime, China). For 500 μL lysate, add 10 mM EDTA (pH 8.0), and 0.1 mM GTPγS or 1 mM GDP. The mixture was incubated at 30 °C for 15 min with constant agitation, then 60 mM MgCl$_2$ was added and the sample was vortexed. 100 μL 50% resin slurry was added to the spin cup with a collection tube, wash with 400 μL 1× Lysis/Binding/Wash buffer to each spin cup with resin, centrifuge the tubes at 6000×$g$ for 10–30 s. and discard the flow-through. The spin cup containing glutathione resin was added 20 μg GST-PAK1-PBD. Immediately, transfer at least 500 μg total protein to the spin cup and incubated the reaction mixture at 4 °C for 60 min with gentle rocking, centrifuge the spin cup with a collection tube at 6000×$g$ for 10–30 s. Wash the resin three times by 1× Cell Lysis/Binding/Wash buffer. Add 50 μL 2× reducing sample buffer with 200 mM DTT to the resin and incubate at room temperature for 2 min. Finally, the binding proteins were collected by centrifuging the tube at 6000×$g$ for 2 min, then denaturing the eluted samples for 5 min at 95–100 °C. Samples may be electrophoresed on a gel or stored at −80 °C for future use.

## Animal experiments

Four-week-old female nude mice were purchased from Vital River Company (Beijing, China) and maintained in Specific Pathogen Free (SPF) animal facility (68–71.6 °F temperature and 50–60% humidity). 8 × 10^5 MGC803 cells constitutively expressing luciferase cells in 0.2 ml PBS were injected into the tail vein of mice. Ten days later, pentobarbital-anesthetized mice were injected intraperitoneally with the D-luciferin substrate (PerkinElmer, USA; 15 mg/ml in PBS) and imaged under a Bruker In-Vivo Xtreme system to observe the metastasis. For all analyses, an additional ROI was employed to normalize for background luminescence on each image. After the metastasis model was established in most mice, they were randomly divided into three groups (6 mice per group). **14-5-18** was dissolved in a mixed solvent of DMSO (2%) and Tween 20 (5%) in water, and administrated by gavage at 0, 10, and 30 mg/kg for each group, once a day. After 18 days of administration, the mice were imaged with a Bruker In-Vivo Xtreme system. Mice were sacrificed, and their lungs were removed and stained by Hematoxylin and Eosin (H & E) staining.

## Construction of virtual combinatorial compound library

The literature on the organic synthetic methodology of our group from 2008 was collected, including 144 papers. Firstly, the substituent libraries were established use Sybyl-X 2.0 to build various types of substituent libraries, such as alkyl substituent library, aromatic ring substituent library, etc. Then, the compound skeletons with marked derivable sites based on the reported literatures was formatted as MOL2 for the subsequent virtual combination. According to reported

substrate scope of each reaction, the proper substituent libraries were selected for each derivable site to ensure the feasibility of synthesis. Finally, the Legion Module in Sybyl-X 2.0 was used to construct the virtual combinatorial compound library.

### Similarity comparison

The similarity comparison of SMBL-E and SMBL-V versus the commercial libraries (*chembridge*, *targetmol*, and *specs*) was performed using 2D fingerprint Tanimoto coefficient (Tc) calculations in Sybyl-X 2.0, the "compare databases" function of the Library Design module.

### Molecular docking

The structure of GIT1 was downloaded from Alpha fold (https://alphafold.com/entry/Q9Y2X7). The binding interaction of GIT1/β-PIX was modeled according to the crystal structure of GIT2/β-PIX (PDB ID: 6JMT). The binding domain of GIT1 protein with amino acids from 8 to 357 was extracted for further study. The structure was prepared by Sybyl-X 2.0 software (Tripos Associates, St. Louis, MO, USA) with the Powell method under AMBER7 FF99 force field and AMBER charges. Protonation states of ionizable residues and histidine residues were predicted according to the microenvironment and $pK_a$ values calculated by the PDB2PQR Server 13,14 at pH = 7.0. The binding pocket of GIT1 was generated by scanning the muti-channel surface, and the channel surface include the amino acid of Leu271 and Leu279 was selected to generate the binding Protomol file. The initial virtual screening was carried out using surflex-dock_cluster as command:

surflex-dock_cluster.sh -o "-multistart 0 -self_score -ring -rigid +soft_box +premin +remin +frag +spinalign -spindense 3.00" -d ligands.mol2 -T "protomol.mol2 receptor.mol2" -n 20 -w 30 -N redocking-6A -r "+rescorefast +dumpprot +pflex" -R PF.

A further docking study was performed using Surflex-Dock in Sybyl-X 2.0 software with Surflex-Dock Geomx (SFXC) mode. The pre-dock minimization, post-dock minimization, consider ring flexibility, molecule fragmentation, and soft grid treatment was set as on. The binding pocket was generated based on the key residues with default setting (Threshold 0.5 and Bloat 0).

On the other side, a further docking study was performed using Autodock. In this calculation, the free open-source package Autodock41 (AD4) and Autodock Vina2 (Vina) were employed to calculate the ligand-binding affinity. The experimental pose was selected docking site ($x = -20.86$, $y = 21.17$, $z = -147.94$) and the grid size was set to $25\,Å × 25\,Å × 25\,Å$, which is a large enough cavity to cover the entire target active site. The largest ligand-binding affinity was the best docking results. Moreover, 6272 ($196 × 32$) ligand poses were recorded and the performance was evaluated by the Vina approach.

### Molecular dynamics simulation

The docked complex with the highest score was chosen for the molecular dynamic simulation. All MD simulations were carried out using AMBER14 with ff14SB force field for protein and gaff force field for small molecules. The structures were prepared with the *tleap* module and minimized with *pmemd.MPI*. The small molecules were treated by antechamber and Gaussian 09 programs. The MD simulations were run with *pmemd.cuda.MPI* executable using Graphical Units Processors module. The systems were neutralized with $Na^+$ or $Cl^-$ firstly, then solvated in the TIP3P water model and subsequently placed into a regular hexahedron box with a minimal distance of 12 Å for the solute from the box borders. After minimization and equilibration, MD simulations for the different systems were performed, respectively. MD simulations were run under periodic boundary conditions using NPT ensemble at 300 K.

### Trajectory analysis

The simulation trajectories were analyzed using the *cpptraj* module of Amber 14. The root means square deviation (RMSD) was calculated.

The equilibrium of the system was determined according to RMSD values. From MD simulation times when the protein reached equilibrium, the average structures of the models were calculated using the *cpptraj* module.

### Calculation of binding free energies

To calculate the binding free energies of proteins with their respective ligands, MD simulations were performed using the aforesaid MD protocol, until the systems reached equilibrium. The binding free energies were calculated using the MM/GBSA method implemented in AMBER. Totally 100 snapshots were extracted from the equilibrium trajectory for MM/GBSA free energy calculation. Per residue energy decomposition was also performed to evaluate the energy contribution of each residue in the systems. All the other parameters were kept as default value. Based on the binding free energies calculation, we identified the key residues that employing more contribution to the binding interaction.

### Statistical analysis

The statistical analysis was performed using Graphpad Prism 8.0 software. Otherwise noted, data were presented as mean ± SD from a minimum of three independent experiments. For numerical results, two-tailed $t$ tests and one-way ANOVA (Dunnett's multiple-comparisons test for each group compared with the control group, and Tukey's multiple-comparisons test for each group compared with every other group) were used to compare the difference between two or multiple groups, respectively. Two-tailed Fisher's exact test was used to compare the proportion of samples. Exact $P$ values are presented in the graphs where applicable, except when values <0.0001 in one-way ANOVA multiple analysis or <0.000001 in $t$ tests, which could not be generated using Graphpad Prism 8.0. Statistical significance was established when $p < 0.05$.

### Reporting summary

Further information on research design is available in the Nature Portfolio Reporting Summary linked to this article.

## Data availability

All data generated in this study are provided in the Source Data file. The protein data of GIT2/β-Pix used in this study are available in the PDB database under accession code 6JMT. The GIT1 structure is available in Alpha Fold under accession code Q9Y2X7. The information on the tissue microarray of human gastric carcinoma used for IHC could be accessed on the website of the company under chip No. HStmA180Su11 [http://outdobiotech.com/tissue.html]. Source data are provided with this paper.

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

## Acknowledgements

We sincerely thank our collaborators Prof. Ying-Chun Chen and Prof.Wei Du at West China School of Pharmacy, Sichuan University for contributing the compounds to establish the library. This work was partially supported by a grant from the National Key R&D Program of China (2018YFA0507900 to Q.O. and J.W.Z.), a grant from National Science and Technology Innovation 2030 Major Projects for "Brain Science and Brain-Inspired Research" (2022ZD0214400 to J.W.Z.), grants from National Natural Science Foundation of China (22007100 to J.G.; 82273775 to Q.O.; 32122036, U2032122 to J.W.Z.), a grant from Science and Technology Commission of Shanghai Municipality (20S11900200 to J.W.Z.).

## Author contributions

Q.O. and J.W.Z. conceived and supervised the overall project. J.G. designed the biological experiments. R.K.P. and C.L.G. performed all the cellular and animal experiments and analyzed the data. J.Y. and X.Y. contributed to the virtual library construction and virtual screening, under the direction of Q.O. Q.Z. and H.L. implemented the FP experiments to evaluate the compounds. N.W. took part in the chemical synthesizing. M.Z. and J.W.Z. provided the purified proteins and designed the mutations. J.G. and R.K.P. did the data collection and prepared the figures. Q.O., J.W.Z. and J.G. drafted the manuscript with critical feedback from all authors.

## Competing interests

The authors declare no competing interests.
