## [Peer Review File Final · Nature Communications]

Construction of A Synthetic Methodology-Based Library and Its Application in Identifying a GIT/PIX Protein–Protein Interaction InhibitorREVIEWER COMMENTS

Reviewer #1 (Remarks to the Author):

The manuscript 'Construction of a synthetic methodology-based library and its application in identifying the first GIT/PIX protein-protein interaction inhibitor' by Gu et al. reports an interesting approach for library design and the successful screening for an disruptor of a hitherto not successfully targeted protein protein interaction.

The paper starts with building of a historical library of compounds from previous synthetic projects of the group between 2012 and the date of the completion of the manuscript. This library is actually not exceedingly big, with around 1,600 compounds produced over roughly 10 years. However, to keep such a library in good shape is an achievement for an academic group. In any case, this physical library is complemented by a substantially larger virtual library (14 million compounds).

These libraries were used to identify inhibitors of the GIT/PIX protein-protein complex, which plays an important role in gastric cancer. A crystal structure of this complex solved by the group was used for the in-silico identification of disruptors of this complex. In addition, a biochemical screen (FP) was used to find such inhibitors from the physical library.

The latter is somewhat confusingly described in the text as it states '...Several representative compounds (N=100) were chosen....Of which 109 compounds were subjected to the FP binding assay...'

I have 2 questions on this:

1. How do you choose 109 compounds from a group of 100?
2. Why hasn't the entire library (1,600) been screened? That shouldn't be any problem with a FP assay

Two compounds were identified from these screening efforts and tested in cell systems and a mouse model for activity, where one of these compounds showed promising activity.

For my taste, the step from a screening hit from a biochemical assay and supported by a docking to the proposed binding target (GIT1) is a bit fast. I would have liked to have seen some independent biophysical proof of the physical interaction of the compound with the target protein. On the other hand, the pull-down data as well as the mutational study are both quite convincing. Nonetheless, showing the direct binding of the compound to GIT1 for example by ITC, NMR or X-ray crystallography would strongly increase the value of the work.

All in all, this is a very nice and valuable work to show the use of a newly established library in targeting a difficult PPI. I believe that this manuscript will be received positively by the readership of the journal.

Minor remarks:

1. Abstract, line 35: I think 'illustrates' fits better than 'illustrated'.
2. Caption Fig.2, line 231: could the PDB code for the structure shown in c be given?

Reviewer #2 (Remarks to the Author):

In this study, Gu et al. describe the collation and application of a synthetically-attainable, natural-product-like, diverse chemical library. They outline the novelty of a curated physical library of compounds, and its expansion as a virtual library for virtual screening with the advantage of in-built synthetic attainability. The increased chemical diversity of this library is proposed to increase the likelihood of finding chemical starting points to modulate novel targets and biology, with a focus on protein-protein interactions. The identification of hit compounds against such challenging and novel

targets is of significant and growing interest to the broader drug discovery community.

The authors applied this compound set to a screen against a PPI between GIT1 and β -Pix. Hit compounds were identified from both unbiased screening of the physical collection and selected compounds tested from parallel virtual screening, as measured in a protein-based peptide competition assay using mouse GIT1 and a peptide from β -Pix.

The hits are subsequently evaluated in a range of cell-based assays using gastric cancer cell lines, before demonstrating in vivo activity in a model of gastric cancer metastasis. While the translation to in vivo models strengthens the clinical relevance of the study, the key data relates to demonstrating the desired mode of action of the hit compounds in in vitro systems to support the applicability of the described libraries for identifying on-target PPI inhibitors.

The authors present co-IP studies to demonstrate reduction of the GIT1/ β -Pix PPI in cells, and immunofluorescence supporting reduced GIT1/ β -Pix co-localisation. CETSA is used to measure compound target engagement with GIT1 in cell lysates and modulation of proximal biomarkers is shown through reduction in cellular levels of GTP-Rac1 and GTP-Cdc42. The compounds are shown to ablate the invasiveness of gastric cancer cell lines in vitro. Mutation of residues suspected to be important for compound binding, as determined from in silico docking, is explored to strengthen the case for on-target activity.

While not entirely conclusive, collectively these datasets support an on-target mechanism of the hit compounds identified. However there are some areas where the supporting data could be strengthened. As the authors describe, the identified hits show weak activity, requiring careful interpretation of cell-based data, much of which is at concentrations of 20 – 100 μ M. The manuscript should seek to strengthen through both increased replicates, and alternative endpoints, the data ruling out generic off-target effects, and some data indicating on-target effects.

Comments:

- There appears to be duplication of some of the co-IP experiments. Fig2a and Fig3a appear to show the same set of co-IPs, with text suggesting data is missing from Fig2a.

- The study uses an in vitro cell invasion assay to support many of its claims. Given the high concentrations of compound used in these studies it is important to effectively rule out generic cytotoxic effects reducing cell numbers regardless of invasion and motility. The authors show data relating to this in Supp Fig4a, and the inactivity of 14-5-18 at 50 μ M following knockdown of GIT1 in Supp Fig 6 also support this, but the manuscript would benefit from strengthening this data. The authors should:

Describe the number of replicates for the current CCK-8 viability data in Supp Fig 4b/c.

A suggested improvement is to add to this data with generic cell viability assessed in MGC803 and MKN45 cells treated with relevant concentrations (covering 10-50 μ M) of 14-5-18 and 15-3-26 for 24 hours by another endpoint, such as quantification of ATP e.g. ATPLite or CellTitreGlo.

- Cell lysate CETSA is used to demonstrate target engagement of hit compounds with GIT1. These studies are challenging due to the weak compound potencies requiring high μ M concentrations, and the apparent high melting temperature of GIT1. The authors demonstrate some evidence for a thermal stabilisation of GIT1. This data would be significantly strengthened through increased replicates of the study.

For the CETSA melt shown in Sup Fig 3c, repeats to n=3 would be beneficial, especially given the DMSO control melt profile for the GIT1 mutants shown in Sup Fig 8 look more similar to the 'compound stabilised' melt profile in Fig 3c.

Additional CETSA melt curves should also seek to analyse the DMSO control and compound treated samples for each temperature side by side on the same blot for more accurate comparison.

- **The dose response CETSA data in Sup Fig 3b strengthens this data set. The authors should clarify the number of replicates from the quantified data which shows error bars, as only n=1 blot is shown in the figure.**

The figure legend (65°C) and methods (60°C) do not match in terms of the heat shock temperature used in these experiments.

- **The study contains a number of experiments using GIT1 mutations of Leucine residues proposed to be essential for the GIT/ β -Pix PPI and to contribute to the ligand binding pocket.**

The authors state the mutation of these residues to Ala "largely abolished the GIT1/ β -Pix interaction" (line 180), yet the data in Fig 2a does not show this and does not appear to be in the manuscript. Fig2a appears a duplication of Fig 3a.

Fig4c measures GTP-Rac biomarker response +/- the test compound where biomarker modulation (reduced GTP-Rac) is shown with re-introduced WT GIT1 but not with GIT1 mutants. However, the compound appears to cause the biomarker response in the absence of any expressed GIT and in a GIT1 knockdown background. Can the authors comment why the compound is reducing GTP-Rac in a GIT1 knockdown setting, as this wouldn't appear to support a GIT1 driven effect.

Reviewer #3 (Remarks to the Author):

This interesting study by Gu et al has established a synthetic methodology-based compound library (SMBL) and further selected two compounds 14-5-18 and 15-4-26 to test their inhibitory effect on GIT1/ β -Pix interaction and their potential application in gastric cancer metastasis. In general, most experiments are well designed and carefully conducted, providing interesting information about synthetic methodology. Most results are consistent and support the conclusions that compound 14-5-18 inhibited GIT1/ β -Pix interaction and cancer cell invasion, whereas compound 15-4-26 exhibited weaker inhibitory effect. Mechanistic studies suggest that 14-5-18 suppressed gastric cancer metastasis by targeting residues Leu271 and Leu279 within GIT1. However, overall novelty of this study needs to be improved, and there are also several specific concerns with the current manuscript.

Specific Concerns:

- 1. The confocal images presented in Fig. 3 are representatives of cellular distribution of GIT1 and β -Pix with or without 14-5-18 treatment. However, it seems that there were only small percentages of cells that were reacted to this treatment. It would be helpful if additional analysis were provided to quantitate changes of cellular distribution of GIT1 and β -Pix.**
- 2. The in vivo study showed that 14-5-18 effectively reduced gastric cancer metastasis. However, it is unclear why the measurements were limited only in tumor size and number after administration of 14-5-18 and how the time points were selected in this study. There were not enough results indicating that observed effect of 14-5-18 was mediated through inhibition of GIT1/ β -Pix interaction.**
- 3. Knock-down cell line (MKN45-shGIT1 cells) was used to study the mechanism underlying inhibitory effect of 14-5-18. But it is necessary to provide results to confirm efficiency of the inhibition. There are similar concerns with studies using transfection of GIT1 mutants.**

- 4. It would be helpful there were the solvent control when the compounds were tested in vitro as well as in vivo.**
- 5. It would be better if Results in the "Figures" were grouped according to summarized "results".**
- 6. The results (line 173-192) are about GITImuts and how it could abolish the GITI/ β -Pix interaction. It seems these results are not properly presented in the referred Fig 2a and 3b.**

Re: Manuscript ID: NCOMMS-22-15144A

1. Reviewer #1 (Remarks to the Author):

The manuscript 'Construction of a synthetic methodology-based library and its application in identifying the first GIT/PIX protein-protein interaction inhibitor' by Gu et al. reports an interesting approach for library design and the successful screening for an disruptor of a hitherto not successfully targeted protein protein interaction.

The paper starts with building of a historical library of compounds from previous synthetic projects of the group between 2012 and the date of the completion of the manuscript. This library is actually not exceedingly big, with around 1,600 compounds produced over roughly 10 years. However, to keep such a library in good shape is an achievement for an academic group. In any case, this physical library is complemented by a substantially larger virtual library (14 million compounds).

Response: We sincerely thank the reviewer for his/her encouraging feedback and positive comments on our work.

● These libraries were used to identify inhibitors of the GIT/PIX protein-protein complex, which plays an important role in gastric cancer. A crystal structure of this complex solved by the group was used for the in-silico identification of disruptors of this complex. In addition, a biochemical screen (FP) was used to find such inhibitors from the physical library.

The latter is somewhat confusingly described in the text as it states '...Several representative compounds (N=100) were chosen...Of which 109 compounds were subjected to the FP binding assay...'

I have 2 questions on this:

1. How do you choose 109 compounds from a group of 100?

2. Why hasn't the entire library (1,600) been screened? That shouldn't be any problem with a FP assay

Response: We are sorry for the unclear description that leads to your confusion. The 109 compounds subjected to FP assay were consisted of two parts: one is 100 representative compounds chosen from the entity library, while the other 9 compounds are from in-silico screening of the virtual library and synthesized thereby. This workflow has been amended in the revised Fig. 2e (see below Figure R1 for convenience). The respective description in the manuscript has been revised to: "Together with the 9 compounds selected from virtual screening, a total of 109 compounds were subjected to the FP binding assay to perform the final evaluation."

Figure R1 (Revised Fig. 2e) Workflow of screening in both entity and virtual libraries to identify **14-5-18** and **15-4-26** as potential inhibitors.

We did not screen the entire library considering the property of the library: most compounds belong to a series of derivatives with same core scaffolds. Because it is generally believed that derivatives with same core structures possess similar activities, we randomly chose 1~3 representatives for each scaffold. For example, the library contains 21 compounds which share same spiro[bicyclo[2.2.1]heptane-2,3'-indolin]-2'-one scaffold with **14-5-18** (shown in Figure R2 below), we chose **14-4-4**, **14-5-18**, and **14-6-18** as representatives for this series of compounds to perform FP assays.

For the selection of representative compounds from the virtual library, 9 compounds from 20 top ranked molecules in docking experiments (one molecule for each scaffold) were synthesized to perform further FP assay (see Supplementary Table 3 for more details). We have clarified the selection principle in the revised manuscript.

Figure R2 The series of compounds with spiro[bicyclo[2.2.1]heptane-2,3'-indolin]-2'-one scaffold. A total of 21 compounds were shown and **14-4-4**, **14-5-18**, and **14-6-18** were selected as representatives for the screening.

- *Two compounds were identified from these screening efforts and tested in cell systems and a mouse model for activity, where one of these compounds showed promising activity. For my taste, the step from a screening hit from a biochemical assay and supported by a docking to the proposed binding target (GIT1) is a bit fast. I would have liked to have seen some independent biophysical proof of the physical interaction of the compound with the target protein. On the other hand, the pull-down data as well as the mutational study are both quite convincing. Nonetheless, showing the direct binding of the compound to GIT1 for example by ITC, NMR or X-ray crystallography would strongly increase the value of the work.*

Response: We fully agree with the reviewer that showing the direct binding of the compound to GIT1 would further increase the value of our work. Following the reviewer's wonderful suggestion,

we have determined the binding affinity between **14-5-18** and GIT1 using the biolayer interferometry (BLI) assay which has been widely utilized in evaluating the bindings of molecules to proteins (1-3). The BLI experiment showed that the compound **14-5-18** bound to GIT1 with a K_D value of $7.7 \pm 0.1 \mu\text{M}$ (Figure R3 below). We have included this result in the revised manuscript (revised Fig. 3a).

Figure R3 (Revised Fig. 3a) Biolayer interferometry (BLI) sensorgram demonstrating the binding kinetics of compound **14-5-18** to GIT1.

1. Vogel, A. B. et al. BNT162b vaccines protect rhesus macaques from SARS-CoV-2. *Nature* **592**, 283 (2021).
2. Brouwer, P. J. M. et al. Two-component spike nanoparticle vaccine protects macaques from SARS-CoV-2 infection. *Cell* **184**, 1188–1200 (2021).
3. ANITA C. BELLAIL Bellail, A. C. et al. Ubiquitination and degradation of SUMO1 by small-molecule degraders extends survival of mice with patient-derived tumors. *Sci. Transl. Med.* **13**, eabh1486 (2021).

● *Minor remarks:*

1. Abstract, line 35: I think 'illustrates' fits better than 'illustrated'.

Response: Revised as suggested with thanks.

2. Caption Fig.2, line 231: could the PDB code for the structure shown in c be given?

Response: The PDB code for the structure is 6JMT. We have included the information in the revised manuscript.

2. Reviewer #2 (Remarks to the Author):

In this study, Gu et al. describe the collation and application of a synthetically-attainable, natural-product-like, diverse chemical library. They outline the novelty of a curated physical library of compounds, and its expansion as a virtual library for virtual screening with the advantage of in-built synthetic attainability. The increased chemical diversity of this library is proposed to increase the likelihood of finding chemical starting points to modulate novel targets and biology, with a focus on protein-protein interactions. The identification of hit compounds against such challenging and novel targets is of significant and growing interest to the broader drug discovery community.

The authors applied this compound set to a screen against a PPI between GIT1 and β -Pix. Hit compounds were identified from both unbiased screening of the physical collection and selected compounds tested from parallel virtual screening, as measured in a protein-based peptide competition assay using mouse GIT1 and a peptide from β -Pix.

The hits are subsequently evaluated in a range of cell-based assays using gastric cancer cell lines, before demonstrating in vivo activity in a model of gastric cancer metastasis. While the translation to in vivo models strengthens the clinical relevance of the study, the key data relates to demonstrating the desired mode of action of the hit compounds in in vitro systems to support the applicability of the described libraries for identifying on-target PPI inhibitors.

The authors present co-IP studies to demonstrate reduction of the GIT1/ β -Pix PPI in cells, and immunofluorescence supporting reduced GIT1/ β -Pix co-localisation. CETSA is used to measure compound target engagement with GIT1 in cell lysates and modulation of proximal biomarkers is shown through reduction in cellular levels of GTP-Rac1 and GTP-Cdc42. The compounds are shown to ablate the invasiveness of gastric cancer cell lines in vitro. Mutation of residues suspected to be important for compound binding, as determined from in silico docking, is explored to strengthen the case for on-target activity.

Response: We sincerely thank the reviewer for the careful reading of our manuscript. The reviewer brings in an excellent remark.

● *While not entirely conclusive, collectively these datasets support an on-target mechanism of the hit compounds identified. However there are some areas where the supporting data could be strengthened. As the authors describe, the identified hits show weak activity, requiring careful interpretation of cell-based data, much of which is at concentrations of 20 – 100 μ M. The manuscript should seek to strengthen through both increased replicates, and alternative endpoints, the data ruling out generic off-target effects, and some data indicating on-target effects.*

Response: We appreciate the reviewer's concern with the on-target effect of the hit compounds. Following the wonderful suggestions from the reviewer, we have increased replicates of the CETSA assay and applied the suggested CellTiterGlo assay to assess generic cell viability of gastric cancer cells treated by hit compounds.

In addition, to obtain more solid evidence to indicate direct binding of compound **14-5-18** to GIT1, we have performed the biolayer interferometry (BLI) experiment to measure the binding affinity of the interaction. The BLI experiment showed that the compound **14-5-18** bound to GIT1 with a K_D value of $7.7 \pm 0.1 \mu$ M (Figure R4 below, see Revised Fig. 3a). Together with the knock-down and mutation experiments, we hope that these data would be convincing to indicate the on-target effects.

Figure R4 (Revised Fig. 3a) Biolayer interferometry (BLI) sensorgram demonstrating the binding kinetics of compound **14-5-18** to GIT1

Comments:

- There appears to be duplication of some of the co-IP experiments. Fig2a and Fig3a appear to show the same set of co-IPs, with text suggesting data is missing from Fig2a.

Response: We thank the reviewer to point this out and we apologize for the careless mistake. Fig. 2a has been corrected in our revised manuscript. For convenience, we have included the result below as Figure R5.

Figure R5 (Revised Fig. 2a) Co-immunoprecipitation (co-IP) experiments to detect the interaction between β-Pix and GIT1 in both the wild-type or mutant forms. HA-GIT1 (wide-type and mutant forms) and Flag-β-Pix were transfected into HEK-293T and the cell lysates were collected to perform the co-IP assay.

- The study uses an *in vitro* cell invasion assay to support many of its claims. Given the high concentrations of compound used in these studies it is important to effectively rule out generic cytotoxic effects reducing cell numbers regardless of invasion and motility. The authors show data relating to this in Supp Fig4a, and the inactivity of 14-5-18 at 50 uM following knockdown of GIT1 in Supp Fig 6 also support this, but the manuscript would benefit from strengthening this data. The authors should:

Describe the number of replicates for the current CCK-8 viability data in Supp Fig 4b/c.

Response: The number of replicates (n = 6) for CCK-8 viability data in Supplementary Fig 4b and c has been added in the figure caption in the revised version of manuscript.

- A suggested improvement is to add to this data with generic cell viability assessed in MGC803 and MKN45 cells treated with relevant concentrations (covering 10-50 μM) of 14-5-18 and 15-3-26 for 24 hours by another endpoint, such as quantification of ATP e.g. ATPLite or CellTiterGlo.

Response: Following the suggestion from the reviewer, we have performed the CellTiterGlo assay to detect the ATP level of MGC803 and MKN45 cells treated with 14-5-18 and 15-4-26. As shown in Figure R6 below, the two compounds did not significantly affect the ATP level of the two cell lines. These results together with our CCK8 experimental results indicated that treatment of hit compounds had limited effect on the generic cell viability of MGC803 and MKN45 cells. Therefore, we reasonably conclude that the reduced invasion ability of cells treated with hit compounds was mainly attributed to on-target effect of these compounds. We have included these new data in the revised Supplementary Fig. 4d and 4e.

Figure R6 (Revised Supplementary Fig. 4d & e) CellTiterGlo assay was performed to measure the ATP level of the MGC803 (d) and MKN45 (e) cell line treated with 14-5-18 and 15-4-26 at 1.25, 2.5, 5, 10, 20, 40, 80, 100 μM for 24 h. Noted that the ATP level of two cell lines was not obviously influenced even when the concentration was increased to 100 μM .

- Cell lysate CETSA is used to demonstrate target engagement of hit compounds with GIT1. These studies are challenging due to the weak compound potencies requiring high μM concentrations, and the apparent high melting temperature of GIT1. The authors demonstrate some evidence for a thermal stabilisation of GIT1. This data would be significantly strengthened through increased replicates of the study.

For the CETSA melt shown in Sup Fig 3c, repeats to $n=3$ would be beneficial, especially given the DMSO control melt profile for the GIT1 mutants shown in Sup Fig 8 look more similar to the 'compound stabilised' melt profile in Fig 3c.

Additional CETSA melt curves should also seek to analyse the DMSO control and compound treated samples for each temperature side by side on the same blot for more accurate comparison.

Response: We totally agree with the reviewer that the CETSA experiments are challenging in this case because of the moderate binding affinity of the compound and the high melting temperature of GIT1. Nevertheless, as suggested by the reviewer, we have repeated the CETSA experiments ($n=3$) using wild-type and mutant form of GIT1 treated with 14-5-18 as well as the DMSO control to evaluate target engagement of hit compound. As shown in Figure R7 below, the wide-type GIT1 shows a rapid melting between 60-70 $^{\circ}\text{C}$ in the DMSO control group, and compound 14-5-18 displayed obvious thermal stabilizing effect on GIT1 over 70 $^{\circ}\text{C}$, which partially demonstrated target engagement of hit compound with GIT1. We have updated the text and figures in the revised

manuscript (see Revised Supplementary Fig. 3c). However, we did not obtain repeated convincing CETSA data using the GIT1 mutants most likely because these mutate proteins were unstable in solution. To keep data reliable, we decide to remove the results involving the CETSA experiments of GIT1 mutants. We believe this change does not influence the overall significance of work. Importantly, in order to show evidence of the direct interaction between compound **14-5-18** and GIT1, we have determined the binding affinity of the interaction through the biolayer interferometry (BLI) experiment (see Figure R4 above for detail). Taken together, we confidently believe that compound **14-5-18** directly binds to GIT1.

Figure R7 (Revised Supplementary Fig. 3c) CETSA assays to detect the interaction between **14-5-18 with GIT1.** Immunoblotting of cell lysates was used to measure the stability of wide-type GIT1, when adding **14-5-18** at 50 μM under 30, 40, 50, 60, 70, 80, 90 $^{\circ}\text{C}$. The graph shows the quantified results.

- *The dose response CETSA data in Sup Fig 3b strengthens this data set.*
The authors should clarify the number of replicates from the quantified data which shows error bars, as only $n=1$ blot is shown in the figure.
The figure legend (65°C) and methods (60°C) do not match in terms of the heat shock temperature used in these experiments.

Response: Following the suggestion from the reviewer, the dose response CETSA assay has been performed twice at concentrations of 0.5, 1.0, 5.0, 10, 25, 50 and 100 μM of **14-5-18** to evaluate the dose-dependent stabilizing effect of **14-5-18** on GIT1. GIT1 was obviously stabilized by increased concentrations of **14-5-18** (Figure R8 below). We have included these results in the revised Supplementary Fig. 3c. The number of replicates from the quantified data ($n=2$) has also been updated in the supplementary material.

The dose-dependent thermal shift assay was performed at 65°C . We thank the reviewer to point this mistake out and have corrected it in the revised manuscript.

Figure R8 (Revised Supplementary Fig. 3b) CETSA assays to detect the interaction between **14-5-18** with **GIT1**. Immunoblotting of cell lysates was used to measure the stability of **GIT1**, when adding **14-5-18** at 0.5, 1.0, 5.0, 10, 25, 50, 100 μM under 65 °C. The graph shows the quantified results.

● *The study contains a number of experiments using **GIT1** mutations of Leucine residues proposed to be essential for the **GIT/β-Pix** PPI and to contribute to the ligand binding pocket. The authors state the mutation of these residues to Ala “largely abolished the **GIT1/β-Pix** interaction” (line 180), yet the data in Fig 2a does not show this and does not appear to be in the manuscript. Fig2a appears a duplication of Fig 3a.*

Response: We apologize for the careless mistake. Fig. 2a has been corrected in our revised manuscript. For convenience, we have included the result below as Figure R9.

Figure R9 (Revised Fig. 2a) Co-immunoprecipitation (co-IP) experiments to detect the interaction between β-Pix and **GIT1** in both the wild-type or mutant forms. **HA-GIT1** (wide-type and mutant forms) and **Flag-β-Pix** were transfected into HEK-293T and the cell lysates were collected to perform the co-IP assay.

● Fig 4c measures GTP-Rac biomarker response +/- the test compound where biomarker modulation (reduced GTP-Rac) is shown with re-introduced WT GIT1 but not with GIT1 mutants. However, the compound appears to cause the biomarker response in the absence of any expressed GIT and in a GIT1 knockdown background. Can the authors comment why the compound is reducing GTP-Rac in a GIT1 knockdown setting, as this wouldn't appear to support a GIT1 driven effect.

Response: We are sorry for the wrong labeling that caused your confusion. We have corrected the Fig. 4c in the revised manuscript and included it as Figure R10 below for your convenience. The first two lanes of the WB bands represent the shNC group which functions as a control. In this scenario, the MKN45 cells transfected with a scrambled shRNA expression plasmid were treated with or without compound **14-5-18**. As expected, GTP-Rac1 was significant reduced when the MKN45 cells were treated with **14-5-18** (Figure R10), due to impaired endogenous GIT1/ β -Pix interaction caused by **14-5-18** treatment.

Figure R10 (Revised Fig. 4c) The cellular activity of Rac1 in MKN45 cells with wide-type or mutant GIT1 with or without **14-5-18** (50 μ M) treatment for 24 h.

In addition, we have also evaluated the level of GTP-Rac1 in MKN45-shGIT1 cell line with or without **14-5-18** treatment. As shown in Figure R11 below, the drug-induced inhibition of GTP-Rac1 was not observed in MKN45-shGIT1 cell line. Taken together, these data support our conclusion that reduction of GTP-Rac1 and consequent cell invasion ability of drug-treated gastric cells may be mainly attributed to disruption of the GIT1/ β -Pix interaction by hit compounds.

Figure R11 (Revised Supplementary Fig. 6b) The cellular activity of Rac1 in MKN45-shGIT1 cell with or without **14-5-18** (50 μ M) treatment for 24 h.

3. Reviewer #3 (Remarks to the Author):

● This interesting study by Gu et al has established a synthetic methodology-based compound library (SMBL) and further selected two compounds 14-5-18 and 15-4-26 to test their inhibitory effect on GIT1/ β -Pix interaction and their potential application in gastric cancer metastasis. In general, most experiments are well designed and carefully conducted, providing interesting information about synthetic methodology. Most results are consistent and support the conclusions that compound 14-5-18 inhibited GIT1/ β -Pix interaction and cancer cell invasion, whereas compound 15-4-26 exhibited weaker inhibitory effect. Mechanistic studies suggest that 14-5-18 suppressed gastric cancer metastasis by targeting residues Leu271 and Leu279 within GIT1. However, overall novelty of this study needs to be improved, and there are also several specific concerns with the current manuscript.

Response: We thank the reviewer for highlighting our study as interesting. We sincerely appreciate the reviewer's concern with the novelty of this study that we may not clarify clearly in the current manuscript. Therefore, we would like to state the innovation of this work here and include it in the revised manuscript.

Chemical libraries are important for the drug discovery. The diversity, the synthetic accessibility, and the library scale are critical to the speed and quality of small-molecule drug development (*Nature* **2019**, 566, 224). Compound libraries could be mainly classified into four types: 1) million-level entity libraries, which are large, accessible but costly, and usually have low structural complexity; 2) DNA-coding libraries, whose structural diversity is limited due to the limitation of reaction types; 3) virtual combinatorial libraries, which are randomly generated *in silico* thus are lack of synthetic accessibility; 4) natural product libraries with diverse and drug-like chemical space (*Nat. Rev. Drug Discov.* **2015**, 14, 111) but limited sources and complex synthesis. Therefore, the most ideal sources for hit discovery will be a synthetically-accessible, natural product-like, diverse chemical library. In the current work, we have established and reported here such a unique synthetic methodology-based library (SMBL) that might be used to discover special hits.

The advantages of the SMBL includes: 1) the structural complexity and "natural product-like" property of chemicals is unique, which is quite different from the known commercial libraries (Figure R12 below, Fig. 1c & d in manuscript). The uniqueness would offer new chemical space for hit discovery; 2) All chemicals in the virtual library could be synthesized quickly based on well-established synthetic methodologies. The library combines the advantages of natural product libraries, commercial entity and combinatorial libraries. Therefore, we believed that increased chemical diversity of this library would bring more chances to the discovery of new hits for other challenging targets. The identification of hit compounds against such challenging and novel targets is of significant and growing interest to the broader drug discovery community (as acknowledged by Reviewer #2). As far as we known, entity/virtual library based on synthetic methodologies similar to SMBL has not been reported yet.

Figure R12 (Fig. 1c & d) Similarity range between the *SMBL* and commercial libraries.

Importantly, the advantages of *SMBL* in lead discovery have been confirmed in this paper by identifying the first small-molecule inhibitors (**14-5-18** and **15-4-26**) targeting GIT1/ β -Pix. This is a quite challenging protein-protein interaction (PPI) to target with small molecules, as manifested by failure of identification of hit compounds using commercial compound library (Supplementary Table 4). The hit compound **14-5-18** was subsequently evaluated in cell-based assays using gastric cancer cell lines and a model of gastric cancer metastasis, demonstrating the advantages of our *SMBL* in drug discovery for some challenging biological targets.

In fact, the molecule **14-5-18** was from our previous paper (*Nat. Chem.* **2017**, 9, 590), with a complex scaffold - spiro[bicyclo[2.2.1]heptane-2,3'-indolin]-2'-one, which includes continuous chiral centers and spiro-bridged ring systems (Figure R13a). We compared the similarity of **14-5-18** with several popular commercial libraries with similarity score lower than 0.6 (Figure R13b). Additionally, we searched the scaffold of **14-5-18** in *ChEMBL* (<https://www.ebi.ac.uk/chembl/>), a manually curated online database of bioactive molecules including 2,331,700 compounds with distinct scaffolds, and found no similar structure even in such a multifarious database (Figure R13c). These analyses highlight the uniqueness and novelty of our *SMBL* library.

Finally, we hope this work would encourage more chemists in the organic synthetic methodology field to collect their products and use them for hit discovery, promoting the cooperation between organic synthesis and drug discovery. Therefore, we believe that this work should attract attention from a broad spectrum of organic synthetic chemists as well as pharmacologists if being published in this journal after revision according to your helpful guidance. We have included these points in the revised manuscript.

Figure R13 The structural uniqueness of **14-5-18**. **a** The 2D and 3D structure of **14-5-18**. **b** Similarity comparison between **14-5-18** and representative commercial libraries. **c** The substructure search result in *ChemBL*.

Specific Concerns:

- The confocal images presented in Fig. 3 are representatives of cellular distribution of GIT1 and β -Pix with or without **14-5-18** treatment. However, it seems that there were only small percentages of cells that were reacted to this treatment. It would be helpful if additional analysis were provided to quantitate changes of cellular distribution of GIT1 and β -Pix.

Response: We agree with the reviewer that quantitative analysis of cellular distribution of GIT1/ β -Pix complex would strongly improve the quality of our manuscript. Notably, fluorescence intensity profile analysis of GFP-GIT1 (green) and β -Pix (red) demonstrated that the two proteins were colocalized in the membrane region of cells (Figure R14a). Treatment of **14-5-18** significantly decreased the colocalization of GIT1 and β -Pix (Figure R14a). Moreover, we have also quantified the percentage of cells where GIT1 and β -Pix colocalized (Figure R14b). These results have demonstrated a significant decrease in the colocalization of GIT1 and β -Pix upon **14-5-18** treatment. We have included these results in the revised manuscript (Revised Fig. 3c & d).

Figure R14 (Revised Fig. 3c & d) Cell distribution of GIT1 and β -Pix with or without treatment of 14-5-18. GFP-GIT1 (green) and Myc- β -Pix were expressed followed by treatment with a fluorescent secondary antibody (red) in MGC803 cell line to construct the fluorescence system. The cells were observed under confocal microscopy after 14-5-18 was added at concentrations of 0 and 20 μ M. Plot fluorescence intensities of the area marked by the white lines in left panels are shown in the right graphs. Green and red curves represent the GIT1 and β -Pix, respectively. Scale bar, 10 μ m. **b** The percentage of GIT1/ β -Pix colocalized cells with or without treatment of 14-5-18. The analysis was performed in n = 30 cells from three independent experiments. Statistical analysis: Chi-square test.

- *The in vivo study showed that 14-5-18 effectively reduced gastric cancer metastasis. However, it is unclear why the measurements were limited only in tumor size and number after administration of 14-5-18 and how the time points were selected in this study. There were not enough results indicating that observed effect of 14-5-18 was mediated through inhibition of GIT1/ β -Pix interaction.*

Response: We agree with the reviewer that direct comparison of GIT1/ β -Pix interaction in metastatic cancer cells in lung with or without 14-5-18 treatment would be ideal. Many attempts were made to perform immunofluorescence imaging or coimmunoprecipitation of GIT1 and β -Pix, as well as to analyze the level of downstream effectors GTP-Rac1 and GTP-Cdc42. Unfortunately, we failed to obtain these data at this time, which we thought may be due to several reasons: 1) the metastatic foci in the lungs were too small to be completely separated, and the trace amounts of available target proteins were challenging to be detected; 2) active GTP form Rac1 or Cdc42 were transient and not stable enough for analysis.

In the current manuscript, we have proved *in vitro* that 14-5-18 targets GIT1/ β -Pix interaction with good specificity, via the transwell experiment, the co-IP analysis, the detection of GTP-Rac1 under expression of wide-type or mutants of GIT1 in gastric cell lines. *In situ*, the CETSA experiment indicates the target engagement of the compound with GIT1. Moreover, to obtain more direct evidence of interaction between GIT1 and hit compound, we have performed biolayer interferometry (BLI) experiment and the result showed a direct binding of 14-5-18 to GIT1 (revised Fig. 3a, Figure R15 below). Taken together, we reasonably conclude that 14-5-18 exhibits inhibitory effects on gastric cancer metastasis most likely through inhibition of GIT1/ β -Pix interaction. However, due to lack of direct *in vivo* evidence, we would like to discuss the limitation in the revised manuscript.

The time points in the animal experiments (including the starting and endpoints of treatment) were selected considering the fluorescence observation and the weight losing of the animals (Supplementary Fig. 5b).

Figure R15 (Revised Fig. 3a) Bi-layer interferometry (BLI) sensorgram demonstrating the binding kinetics of compound **14-5-18** to GIT1

● Knock-down cell line (MKN45-shGIT1 cells) was used to study the mechanism underlying inhibitory effect of 14-5-18. But it is necessary to provide results to confirm efficiency of the inhibition. There are similar concerns with studies using transfection of GIT1 mutants.

Response: Following the suggestion from the reviewer, the efficiency of GIT1 knockdown was confirmed again via Western blotting analysis. As shown in the Figure R16a below, GIT1 expression was stably inhibited after a lentiviral short hairpin (sh)RNA targeting GIT1 infected cell line MKN45. The GIT1 expression in MGC803 after knockdown was also demonstrated in Supplementary Fig. 1e, the first lane of blots. See Figure R16b below for convenience. Together, the results suggested that the efficiency of GIT1 knockdown was reliable in the main cell lines used in this study. The transfection efficiency of wide-type or mutant GIT1 has also been confirmed (Figure R16c) when performing the detection of the GTP-Rac1 (for Fig. 4c). Expression levels of wide-type or mutant GIT1 are generally equal.

Figure R16 a GIT1 expression levels to show the efficiency of GIT1 knockdown after lentivirus infection of cell line MKN45. **b (Supplementary Fig. 1e)** The highlighted lanes show the efficiency

of GIT1 knockdown after lentivirus infection of cell lines MKN45 and MGC803. **c** Wide-type or mutant GIT1 expression level in MKN45-shGIT1 cell line.

- *It would be helpful there were the solvent control when the compounds were tested in vitro as well as in vivo.*

Response: We thank the reviewer for the comment. Actually, respective solvents were utilized as control in all control groups of this study. Relevant descriptions have been added to the revised manuscript to make it clearer.

- *It would be better if Results in the “Figures” were grouped according to summarized “results”.*

Response: Following the reviewer’s suggestion, we have grouped the Results section into 4 parts according to the flow of 4 main figures in our revised manuscript.

- *The results (line 173-192) are about GIT1mut and how it could abolish the GIT1/β-Pix interaction. It seems these results are not properly presented in the referred Fig 2a and 3b.*

Response: We apologize for the careless mistake. Fig. 2a has been corrected in our revised manuscript. For convenience, we have included the result below as Figure R17.

Figure R17 (Revised Fig. 2a). Co-immunoprecipitation (co-IP) experiments to detect the interaction between β-Pix and GIT1 in both the wild-type or mutant forms. HA-GIT1 (wide-type and mutant forms) and Flag-β-Pix were transfected into HEK-293T and the cell lysates were collected to perform the co-IP assay.

REVIEWERS' COMMENTS

Reviewer #1 (Remarks to the Author):

This revised version of the manuscript addresses adequately all of the points mentioned in my report.

Reviewer #3 (Remarks to the Author):

None